# Automated Random Embedding for Practical Bayesian Optimization with Unknown Effective Dimension

## Abstract

Bayesian optimization is widely employed for optimizing complex black-box functions but struggles with the curse of dimensionality. Random embedding, as a dimension reduction strategy, simplifies tasks that possess the effective dimension by optimizing within a low-dimensional subspace. However, determining the effective dimension of a task in advance remains a significant challenge, which influences the selection of the subspace dimensionality and the optimization performance. Traditional methods use fixed subspace dimensions provided by experts or rely on trial and error to estimate subspace dimensions with many resources consumed. To this end, this paper proposes an automated random embedding for high-dimensional Bayesian optimization with unknown effective dimension, called Dynamic Shared Embedding Bayesian Optimization (DSEBO). DSEBO starts with a low dimension and switches to a higher subspace if the solutions in the current subspace show preliminary convergence. DSEBO dynamically determines the dimension of the next subspace based on the quality of the solutions in different subspaces and shares the queried solutions with the new subspace to achieve a better initialization. Theoretically, we derive a regret bound for DSEBO and demonstrate that DSEBO can better balance approximation and optimization errors. Extensive experiments on functions with dimensionality of varying magnitudes and real-world tasks with unknown effective dimensions reveal that, compared with state-of-the-art methods, alternating optimization across different subspaces results in significant improvements in high-dimensional optimization, both in terms of optimization regret and time.

## 1 Introduction

Optimization (Boyd & Vandenberghe, 2004) is widely used in various fields, such as economics (Mehta & Grosan, 2015), machine learning (Freund & Schapire, 1997; Elsken et al., 2019), and reinforcement learning (Qian & Yu, 2021). It aims to find the global optimal solution $x^*$ of the objective function $f(x)$ in the $D$-dimensional search space $\mathcal{X} \subset \mathbb{R}^D$, formally expressed as $x^* = \mathrm{argmin}_{x \in \mathcal{X} \subset \mathbb{R}^D} f(x)$. However, in black-box optimization scenarios, the expression of the objective function is unknown (Shahriari et al., 2016), where gradient-based optimization techniques are ineffective. In contrast, the derivative-free optimization methods (Zhou et al., 2019; Shahriari et al., 2016; Yu et al., 2025) can handle these complex optimization problems.

Bayesian optimization (BO) (Srinivas et al., 2010; Shahriari et al., 2016; Garnett, 2023) is one of the most valuable derivative-free optimization methods for its excellent performance. However, the curse of dimensionality has persistently remained a critical issue in BO (Shahriari et al., 2016; Wang et al., 2016; Binois & Wycoff, 2022; Santoni et al., 2024). Specifically, as the dimension increases, more evaluations of the objective function are required to adequately explore the solution space, making BO challenging to locate the optimal solution efficiently within finite computational resources.

Random embedding (RE) (Wang et al., 2016; Qian et al., 2016) is a remarkable approach to addressing high-dimensional optimization tasks with effective dimension. By constructing a mapping between a lower-dimensional subspace and the original search space via an embedding matrix, optimization can be performed in the low-dimensional subspace, thus alleviating the scalability issue for BO methods.

This approach exploits the inherent effective dimension of high-dimensional optimization tasks, where the function value is influenced by only a few relevant dimensions. By focusing on these dimensions, random embedding significantly enhances the efficiency and performance of high-dimensional BO within limited resources. However, a critical challenge is how to accurately determine the effective dimension of a function. Selecting an appropriate low-dimensional subspace to perform optimization is crucial for the efficacy of embedding technique: a too-small subspace fails to adequately capture the effective dimension of the function, resulting in a loss of optimal solution, while a too-large subspace diminishes the advantages of the reduced dimensionality provided by random embedding.

However, previously in practice, the dimensionality of the low-dimensional subspace is typically determined via heuristic or trial-and-error approaches. These methods usually require performing multiple optimizations across various subspaces to achieve a more effective solution, which incurs significant computational costs. Currently, there is no direct method to predict the effective dimension required to capture the fundamental characteristics of the target function (Papenmeier et al., 2022).

**Problem.** When using BO with RE for high-dimensional optimization tasks with unknown effective dimension, how to automatically determine the appropriate subspace dimension during optimization?

**Contribution.** To address the aforementioned problem, this paper proposes an effective approach to high-dimensional BO by developing a dynamic shared embedding Bayesian optimization (DSEBO) algorithm, which can automatically expand subspace dimension to handle tasks with unknown effective dimension. The shared embedding technique is introduced to leverage solutions from a low-dimensional subspace within a higher-dimensional subspace, thereby better guiding the initialization of the high-dimensional subspace and accelerating convergence. Based on the technique, DSEBO starts from a lower-dimensional subspace, dynamically determines the next subspace dimension according to the convergence of solutions in different subspaces, and facilitates transitions between various subspaces. The theoretical analysis derives a regret bound for DSEBO, highlighting its ability to balance approximation and optimization errors more effectively than GPUCB. Experiments on high-dimensional synthetic functions and real-world tasks show the effectiveness and superiority of DSEBO. The hyper-parameter experiments illustrate the robustness of DSEBO.

The subsequent sections present the related work and preliminaries, describe the proposed DSEBO method, show the theoretical and empirical results, and conclude the paper.

## 2 RELATED WORK

This section reviews the related work: the high-dimensional optimization algorithms, and the multi-armed bandit (MAB) algorithms used for the subspace dimension selection.

### 2.1 HIGH-DIMENSIONAL OPTIMIZATION ALGORITHM

Embedding-based methods define a low-dimensional effective subspace where function values change dramatically, allowing optimization algorithms to operate within the subspace for lower computational costs. Linear embedding methods map low-dimensional points to high-dimensional space via an embedding matrix. Notable works include REMBO (Wang et al., 2016), a foundational method, and SREBO (Qian et al., 2016), which extends REMBO to sequential settings. HesBO (Nayebi et al., 2019) uses hash matrices for linear mapping, SIRBO (Zhang et al., 2019) employs sliced inverse regression to identify the effective dimension, and ALEBO (Letham et al., 2020) provides a comprehensive analysis of the embedding process. Using nested random subspaces to dynamically increase the optimization space from a low dimensionality to a high dimensionality approaching $D$, BAxUS (Papenmeier et al., 2022) can handle high-dimensional tasks. Nonlinear embedding methods use complex functions to map subspaces to the original space, like VAEBO (Gómez-Bombarelli et al., 2018), which employs unsupervised learning to train variational encoders.

Apart from embedding-based methods, there are high-dimensional optimization algorithms that do not rely on subspace embedding. In the BO field, one method is TuRBO (Eriksson et al., 2019), which handles high-dimensional tasks by dividing the optimization space into smaller regions for local optimization. SAASBO (Eriksson & Jankowiak, 2021) identifies and optimizes only on a few important dimensions. DuMBO Bardou et al. (2024) relaxes additive structure constraints by using decentralized message-passing and a refined acquisition function. MCTS-VS (Song et al., 2022)

employs Monte Carlo tree search to iteratively select a subset of variables and optimize them within a low-dimensional subspace. RDUCB (Ziomek & Bou-Ammar, 2023) uses random tree decompositions to build additive Gaussian process (GP) models, leveraging cycle-free pairwise-dimensional interactions for optimization. Recently, SBO-SE (Xu et al., 2025) introduces a robust initialization strategy for the length-scale of GP kernel, addressing the vanishing gradient in high-dimensional BO. In the field of evolutionary algorithms, LMMAES (Loshchilov et al., 2019) approximates the covariance structure with a small set of evolution paths. DCEM (Amos & Yarats, 2020) introduces a differentiable variant of the cross-entropy method using a smooth top-k operation.

All these methods rely on special assumptions, such as effective dimension. The failure to meet these underlying assumptions can lead to a significant decline in performance.

## 2.2 MULTI-ARMED BANDIT

The problem of selecting subspace dimension can be modeled as an MAB problem. Specifically, different candidate dimensions are treated as different arms in the MAB problem, where the convergence value of the optimized function on these dimensions serves as the reward for each arm.

The MAB problem (Sutton & Barto, 2018; Jin et al., 2022) centers on the trade-off between exploration and exploitation. For instance, Softmax (Sutton & Barto, 2018) employs the exponential probability rule for arm selection. Thompson Sampling (TS) (Jin et al., 2022) is a probabilistic model-based approach that samples arms according to their posterior distributions. Additionally, other algorithms like Extreme, Random, $\epsilon$-Greedy, Expectation, Upper Confidence Bound (UCB), and the UCB-E algorithm, which extends the traditional UCB algorithm, are also widely adopted methods in practice.

## 3 PRELIMINARIES

This section briefly introduces Bayesian optimization (BO), the optimal $\epsilon$-effective dimension, and random embedding to explain the necessary preliminary knowledge and notation.

**Bayesian Optimization.** BO (Shahriari et al., 2016; Garnett, 2023) is a well-known derivative-free optimization method that strategically leverages prior knowledge to guide optimization. BO first constructs a surrogate model of the objective function, typically a Gaussian process, based on the observed dataset $\mathcal{D}$. With the model, BO estimates the posterior distribution and calculates an acquisition function to balance "exploration" and "exploitation" in the search space $\mathcal{X} \subset \mathbb{R}^D$, thus determining the candidate $\boldsymbol{x} \in \mathcal{X}$. A common acquisition function is UCB (Srinivas et al., 2010), defined as $\alpha_{\mathrm{UCB}}(\boldsymbol{x}) = \mu_t(\boldsymbol{x}) + \sqrt{\kappa_{t+1}}\sigma_t(\boldsymbol{x})$, where $\mu_t(\boldsymbol{x})$ estimates the objective function and $\sigma_t(\boldsymbol{x})$ represents model uncertainty. The hyper-parameter $\kappa$ controls the trade-off between "exploration" and "exploitation" for efficient optimization. Details are provided in the Appendix A.

**Random Embedding.** Random embedding (Wang et al., 2016; Nayebi et al., 2019) is a popular subspace embedding method, which can effectively achieve dimensionality reduction when the high-dimensional optimization tasks have effective dimension (i.e., the function value is influenced by only a few relevant dimensions). However, it is challenging to meet the effective dimension assumption in real tasks. In this case, Qian et al. (2016) propose the concept of optimal $\epsilon$-effective dimension to relax the assumption, defined as:

**Definition 3.1.** *[Optimal $\epsilon$-Effective Dimension (Qian et al., 2016)] For any $\epsilon > 0$, a function $f : \mathbb{R}^D \to \mathbb{R}$ is said to have an $\epsilon$-effective subspace $\mathcal{V}_\epsilon$, if there exists a linear subspace $\mathcal{V}_\epsilon \subseteq \mathbb{R}^D$, s.t. for all $\boldsymbol{x} \in \mathbb{R}^D$, we have $|f(\boldsymbol{x}) - f(\boldsymbol{x}_\epsilon)| \leq \epsilon$, where $\boldsymbol{x}_\epsilon \in \mathcal{V}_\epsilon$ is the orthogonal projection of $\boldsymbol{x}$ onto $\mathcal{V}_\epsilon$. Let $\mathbb{V}_\epsilon$ denote the collection of all the $\epsilon$-effective subspaces of $f$, and $dim(\mathcal{V})$ denote the dimension of $\mathcal{V}$. We define the optimal $\epsilon$-effective dimension of $f$ as $d_\epsilon = \min_{\mathcal{V}_\epsilon \in \mathbb{V}_\epsilon} dim(\mathcal{V}_\epsilon)$.*

For a high-dimensional optimization problem $\boldsymbol{x}^* = \mathrm{argmin}_{\boldsymbol{x} \in \mathcal{X} \subset \mathbb{R}^D} f(\boldsymbol{x})$, a random matrix $A \in \mathbb{R}^{D \times d}$ with elements sampled from a Gaussian distribution $\mathcal{N}(0, \sigma^2)$ embeds the $d$-dimensional subspace $\mathcal{Z} \subset \mathbb{R}^d$ into the $D$-dimensional search space $\mathcal{X} \subset \mathbb{R}^D$. The optimizer only needs to optimize $\boldsymbol{z} \in \mathcal{Z}$ in the low-dimensional subspace, and then embed $\boldsymbol{z}$ into $\mathcal{X}$ through the matrix $A$ to get the solution $\boldsymbol{x} = p_\mathcal{X}(A\boldsymbol{z})$. Here, $p_\mathcal{X}(A\boldsymbol{z}) = \mathrm{argmin}_{\boldsymbol{x} \in \mathcal{X}} \|\boldsymbol{x} - A\boldsymbol{z}\|_2$ is to project illegal points (i.e. $A\boldsymbol{z} \notin \mathcal{X}$) back to the $\mathcal{X}$ region. Subsequently, the objective function $f$ is evaluated in the solution $\boldsymbol{x}$, and the sample point $(\boldsymbol{z}, f(p_\mathcal{X}(A\boldsymbol{z})))$ is added to the subspace dataset. The optimizer uses the updated dataset to complete the next optimization iteration. See Appendix A for details.

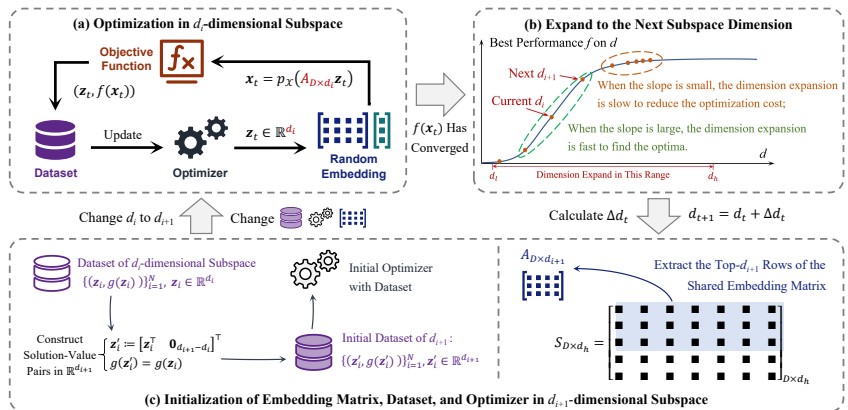

Figure 1: The framework of DSEBO. Subplot (a) shows BO with random embedding, where optimization occurs in a $d_i$-dimensional subspace, and solutions are mapped to the high-dimensional space through random embedding. Subplot (b) shows dynamic dimension expanding, which expands the subspace dimension to achieve an improved evaluation while keeping the dimension not too high. Subplot (c) shows the process of initializing a new subspace, extending the low-dimensional dataset, initializing the optimizer, and sharing the embedding matrix.

## 4 THE PROPOSED METHOD

This section introduces an automated random embedding method, dynamic shared embedding Bayesian optimization (DSEBO), to handle optimization tasks with unknown effective dimension.

### 4.1 DYNAMIC SHARED EMBEDDING BAYESIAN OPTIMIZATION

To automatically identify the appropriate subspace for high-dimensional BO tasks, the DSEBO algorithm is proposed, illustrated in Figure 1 with pseudo-code in Appendix B.

As shown in Figure 1, DSEBO operates in three stages: (a) optimizing in the $d_i$-dimensional subspace, (b) expanding to a new subspace, and (c) initializing the new subspace. DSEBO starts with a lower dimension $d_l$, iteratively optimizes within the subspace until convergence (the convergence criteria will be introduced later), then expands to a higher-dimensional subspace to continue optimization. By gradually increasing and optimizing each subspace, the objective function is optimized while dynamically expanding the subspace dimension. DSEBO considers the differences in the optimal solutions obtained from each subspace when determining dimensionality changes. It also adjusts the scale of these changes during optimization to adapt to functions with different dimensionalities. Besides, DSEBO uses a shared embedding matrix to share evaluation data among different subspaces, allowing data points from low-dimensional subspace to provide better initialization for new high-dimensional subspace, thus conserving the limited budget.

The following section will explain how data are shared between different subspaces and how dynamic strategy is designed to determine the next subspace dimension.

### 4.2 DATASET INITIALIZATION WITH SHARED EMBEDDING

Optimization is performed sequentially on multiple subspaces while dynamically expanding dimensions. However, independent random matrix mappings prevent sharing sampling points across different dimensional subspaces. Therefore, a proposed shared embedding technique enables the utilization of identical sampling points across different subspaces.

The shared embedding technique maintains a shared matrix $S \in \mathbb{R}^{D \times d_h}$, where $D$ is the dimension of the search space and $d_h$ is the dimension of the largest subspace. Each subspace uses a part of this matrix to embed solutions into the search space. Specifically, for a subspace of dimension $d$, the first $d$ rows of the matrix $S$ form the embedding matrix $A_d$. This ensures that for any two subspaces $\mathcal{V}_i$ and $\mathcal{V}_j$ with dimensions $d_i$ and $d_j$ (assuming $d_i < d_j$), $A_{d_i}$ and the first $d_i$ rows of $A_{d_j}$ are identical. Thus, $\mathcal{V}_i$ is a subspace within $\mathcal{V}_j$, allowing $\mathcal{V}_j$ to share sampled data points from $\mathcal{V}_i$.

Figure 2 illustrates the shared embedding process across different subspaces. A 3-dimensional vector $z \in \mathbb{R}^3$ is expanded to 5-dimensional $z' \in \mathbb{R}^5$ by appending zeros. These vectors are then transformed into the search space $\mathbb{R}^D$ using matrices $A_3$ and $A_5$, respectively, from the shared embedding matrix $S$. After embedding, the solutions $x$ and $x'$

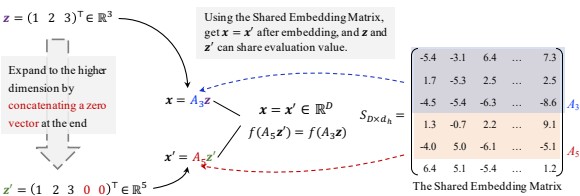

Figure 2: An example of the shared embedding technique.

in the search space are identical, allowing them to share the same evaluation, i.e., $f(A_5 z') = f(A_3 z)$.

Based on the method, each new subspace uses data from the existing lower-dimensional subspace to initialize its dataset. The initialization algorithm is shown in Appendix C. In the first iteration $t = 1$, the dataset is empty and initialized with a random point. For dimension expansion, the current dataset $\mathcal{D}^{(t)}$ is generated from the previous dataset $\mathcal{D}^{(t-1)}$. This is done by appending zeros to each solution vector $z$ to expand it to the new subspace while keeping the original evaluation value $y$.

Through the proposed shared embedding technique, higher-dimensional subspaces can reuse the evaluated data in lower-dimensional subspaces, since the function values of corresponding solutions remain invariant across subspaces due to the connection between subspace embedding matrices.

### 4.3 DYNAMIC DIMENSION EXPANDING STRATEGY

While the shared embedding technique enables new subspaces to reuse existing data from lower-dimensional subspaces, determining optimal switching timing and targets remains a challenge.

Intuitively, the subspace dimension should be updated when the current subspace converges. Specifically, if the optimal value in the subspace remains unchanged for $T$ times, the process can be considered converged. The initial value of $T$ is set to $\lfloor budget/2\beta \rfloor$, controlled by a hyperparameter $\beta$. To determine whether optimization has converged at the current dimension, $T$ is updated as follows: $T = \lfloor (1 + (d^{(t)} - d_l)/(d_h - d_l)) \cdot budget/2\beta \rfloor$, where $d^{(t)}$ is the current subspace dimension, $d_l$ and $d_h$ are the lower and upper bounds of the dimension range. As $d^{(t)}$ increases, so does $T$. This aligns with

---

**Algorithm 1** Dynamic Dimension Expanding Strategy

**Input:** Current dimension $d^{(t)}$, dimension range $[d_l, d_h]$
1: **if** $t \leq 2$ **then**
2: $\quad \Delta d^{(t)} = \lfloor \frac{d_h - d_l}{\beta} \rfloor$
3: **else**
4: $\quad$ Compute the $s_i = -\frac{b_{i+1} - b_i}{d_{i+1} - d_i}, i \in \{1, 2, \dots, n-1\}$
5: $\quad s_{\min}, s_{\max} \leftarrow \min_{i=1}^{n-1} s_i, \max_{i=1}^{n-1} s_i$
6: $\quad$ **if** $s_{\min} = s_{\max}$ **then**
7: $\quad\quad \Delta d^{(t)} \leftarrow \Delta d^{(t-1)}$
8: $\quad$ **else**
9: $\quad\quad \Delta d^{(t)} \leftarrow \lfloor k \Delta d^{(t-1)} \rfloor$, where $k = \frac{s_{n-1} - s_{\min}}{s_{\max} - s_{\min}} + 0.5$
10: $\quad$ **end if**
11: **end if**
12: $d^{(t+1)} = \min(d^{(t)} + \Delta d^{(t)}, d_h)$
**Output:** The next dimension $d^{(t+1)}$.

---

the intuition that higher-dimensional subspaces demand more effort to converge. Based on the definition of $T$, when to update the subspace dimension is solved. Another problem is how much larger the dimension of the new subspace should be.

According to the relationship between subspace dimensions and convergence values, a dynamic dimension expanding strategy is designed to select the next subspace dimension within the specified range $[d_l, d_h]$. Starting from a small dimension $d_l$, the strategy dynamically determines the subsequent dimension, and ultimately identifies a subspace where it can converge to an improved solution without excessively larger dimension. To achieve this, the dimensions of the optimized subspace and the corresponding optimal values are recorded as $d_i$ and $b_i$, where $i$ indicates the $i$-th optimized subspace, with $i \in \{1, 2, \dots, n\}$ representing that $n$ subspaces have been optimized. Based on $d_i$ and $b_i$, the dynamic dimension expanding strategy is designed as Algorithm 1. This strategy determines the next dimension at the $t$-th iteration by calculating the dimension change $\Delta d^{(t)}$. Initially, if fewer than two subspaces have been optimized, $\Delta d^{(t)}$ is set to $\lfloor (d_h - d_l)/\beta \rfloor$, as stated in lines 1–2. Otherwise, the slopes of $b_i$ with respect to $d_i$ are calculated according to: $s_i = -(b_{i+1} - b_i)/(d_{i+1} - d_i)$, where $i \in \{1, 2, \dots, n-1\}$. This equation measures the rate at which the optimal value improves as the

dimensionality of the optimized subspace increases, quantifying the improvement of expanding the subspace. $s_i$ are calculated for minimum optimization and should be inverted for maximum.

Next, a proportion value $k$ is used to determine the change in the current dimension, defined by the formula: $k = (s_{n-1} - s_{\min})/(s_{\max} - s_{\min}) + 0.5$, where $s_{\min} = \min_{i=1}^{n-1} s_i$ and $s_{\max} = \max_{i=1}^{n-1} s_i$. According to the equation, the most recent slope is normalized through min-max scaling to the range $[0.5, 1.5]$ and then used to determine the change in the current dimension. The value of $k$ reflects the impact of increasing the subspace dimension on the convergence value in the current context. If the value of $k$ within the range $[1.0, 1.5]$ indicates a significant impact of the dimension on the convergence value, otherwise it suggests a lesser effect. The algorithm then dynamically scales the change in dimension $\Delta d^{(t-1)}$ from the previous iteration using the value of $k$, yielding the current dimension change $\Delta d^{(t)}$, as described in line 9. Subsequently, based on this dimension change, the new dimension is calculated as $d^{(t+1)} = \min(d^{(t)} + \Delta d^{(t)}, d_h)$, as described in line 12.

We would like to emphasize that, unlike BAxUS (Papenmeier et al., 2022), which increases dimensions exponentially and eventually optimizes in the full space, DSEBO adopts the proposed dynamic dimension expanding strategy. Specifically, DSEBO automatically and nearly linearly expands the subspace dimension within range $[d_l, d_h]$, leading to lower computational costs and better scalability.

## 5 THEORETICAL ANALYSIS

In this section, we present the regret analysis of the naive GPUCB algorithm and DSEBO. To begin, we define $\epsilon(d)$ as the approximation error associated with the $d$-dimensional subspace, formally expressed as $\epsilon(d) = \min_{\boldsymbol{x} \in \mathbb{R}^d} f(\Phi(\boldsymbol{x})) - \min_{\boldsymbol{z} \in \mathbb{R}^D} f(\boldsymbol{z})$, where $\Phi : \mathbb{R}^d \to \mathbb{R}^D$ denotes the embedding that maps a low-dimensional point into the original $D$-dimensional search space. Using this notation, we derive the simple regret bound for the naive GPUCB algorithm.

Here, simple regret is defined as $r_f(T) = \min_{t=1}^T f(\Phi(\boldsymbol{x}_t)) - \min_{\boldsymbol{z} \in \mathbb{R}^D} f(\boldsymbol{z})$, which measures the difference between the best function value found by the algorithm (after embedding the low-dimensional solution back into the high-dimensional space) and the true global optimum.

**Theorem 5.1.** *Suppose that the search space $\mathcal{X}$ is compact and convex with dimension $d$, and every $\boldsymbol{x} \in \mathcal{X}$ satisfies $\|\boldsymbol{x}\|_\infty \leq b$. For GP sample paths $f$ with an RBF kernel, choose $\delta \in (0, 1)$, and define $\beta^{(t)} = 2\log\left(t^2 2\pi^2/3\delta\right) + 2D\log\left(t^2 dbr\sqrt{\log\left(4Da/\delta\right)}\right)$, where $a$ and $b$ are constants satisfying $\Pr\left(\sup_{\boldsymbol{x} \in \mathcal{X}} |\partial f/\partial x_j| > L\right) \leq ae^{-(L/b)^2}$. By running the GPUCB algorithm with $\beta_t$ using an initialization with a zero mean function and a covariance function $k(\boldsymbol{x}, \boldsymbol{x}')$, the simple regret is bounded by $O^*\left(2\epsilon(d) + \sqrt{(\log T)^{d+1}/T}\right)$, where $O^*$ omits logarithmic factors.*

Due to space limitations, we defer the proof in this section to Appendix D. For DSEBO, the dimensions of the subspaces are updated periodically. Let the number of updates be $H$, and denote the subspaces in different phases as $\{\mathcal{Z}_h\}_{h=1}^H$, with dimensions $\{d_h\}_{h=1}^H$ and durations $\{T_h\}_{h=1}^H$. We then present the following theorem. Theorems 5.1 and 5.2 stem from Srinivas et al. (2010); Qian et al. (2016).

**Theorem 5.2.** *Suppose that each subspace $\mathcal{Z}_h$ is compact and convex, and that every $\boldsymbol{x}$ in the subspaces satisfies $\|\boldsymbol{x}\|_\infty \leq b$. For GP sample paths $f$ with RBF kernel, pick $\delta \in (0, 1)$, and define*

$$\beta^{(t)} = 2\log\left(t^2 2\pi^2/3\delta\right) + 2d_h\log\left(t^2 d_h br\sqrt{\log\left(4d_h a/\delta\right)}\right),$$

*where $a$ and $b$ are constants such that $\Pr\left(\sup_{\boldsymbol{x} \in \mathcal{Z}_h} |\partial f/\partial x_j| > L\right) \leq ae^{-(L/b)^2}$. Running DSEBO with $\beta_t$ for the initialization of a GP with mean function zero and covariance function $k(\boldsymbol{x}, \boldsymbol{x}')$, we obtain a simple regret bound of $O^*\left(2\sum_{h=1}^H \epsilon(d_h)T_h/T + \sqrt{\sum_{h=1}^H (\log T_h)^{d_h+1}/T}\right)$.*

**Remarks.** We compare the theoretical results of DSEBO and GPUCB. It can be observed that the first term of DSEBO is larger than that of GPUCB, while the second term is smaller. This indicates that DSEBO effectively balances the trade-off between approximation error and optimization error. To illustrate this, assume that the approximation error satisfies $\epsilon(d') = O^*\left((D - d')/D\right)$. Consider a common setting where $T = 1000$, $H = 10$, $d = 20$, $D = 100$, and the dimension of subspaces increasing evenly up to $d$, substituting these values into the bound yields an approximation error ratio

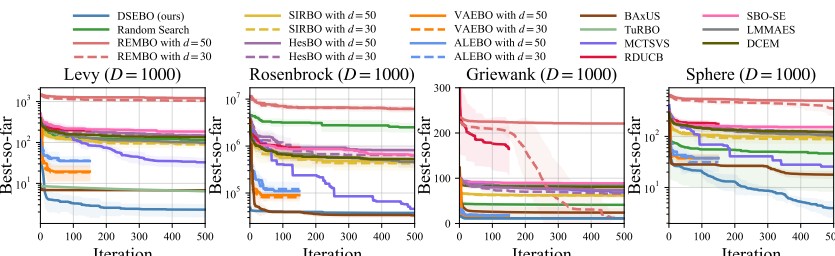

Figure 3: Results on partial synthetic functions compared with various high-dimensional optimization algorithms. All algorithms are independently repeated 10 times.

between DSEBO and GPUCB of $O^*(1.25)$. This implies a modest increase in approximation error for DSEBO compared with GPUCB. However, the optimization error ratio between GPUCB and DSEBO is $O^*(100)$, indicating a significant reduction in optimization error for DSEBO. ***This implies that DSEBO sacrifices a little approximation error in exchange for a much lower optimization error to realize a lower total error (i.e., a better trade-off)***.

# 6 EXPERIMENT

This section evaluates the performance of the proposed DSEBO, shows its effectiveness and superiority through a series of experiments on both synthetic functions and real-world tasks. The code is available in `https://anonymous.4open.science/r/DSEBO-7532`.

**The Setting of Synthetic Functions.** We construct high-dimensional objective functions based on synthetic functions meeting the optimal $\epsilon$-effective dimension (as Definition 3.1). Specifically, let $f : \mathbb{R}^{d_f} \to \mathbb{R}$ be a base testing function, with its domain adjusted to $[-1, 1]^{d_f}$. The high-dimensional synthetic function $F_c : \mathbb{R}^D \to \mathbb{R}$ is crafted to simulate the minimization of $f$, defined by: $F_c(\boldsymbol{x}) = f(\boldsymbol{x}_{1:d_f} - \mathbf{c}) - K^{-1} \sum_{i=d_f+1}^{D} (x_i - c)^2$, where $\boldsymbol{x} \in \mathbb{R}^D$ is the input to $F_c$, and $\boldsymbol{x}_{1:d_f}$ denotes the first $d_f$ dimensions of $\boldsymbol{x}$. The constant vector $\mathbf{c} \in \mathbb{R}^d$, filled with the scalar $c$, is introduced to shift the optimal solution away from the origin. The constant $K$ modulates the influence of dimensions beyond the initial $d_f$. Evidently, $F_c$ possesses an optimal $\epsilon$-effective dimensionality $d_f$, with $\epsilon \leq K^{-1}$. In the experiments, we construct high-dimensional functions with $D = 1000, 10000$ respectively, $d_e = 30$ and $K = 10000$, based on 6 synthetic functions from `http://www.sfu.ca/~ssurjano/optimization.html`, including Levy, Rosenbrock, Griewank, Sphere, and so on. All experiments on synthetic functions are minimum optimization problems.

**The Setting of Real-World Tasks.** We evaluate DSEBO on three real-world datasets. The first dataset is the Microsoft Learning to Rank (MSLR) (Qin et al., 2010), specifically the MSLR-WEB-10K version, containing over 10000 queries, each with 136 features per website page. The second dataset is Lasso-Hard from LassoBench (Sehic et al., 2022), a 1000-dimensional optimization task designed to identify sparse regression coefficients that minimize Lasso regression loss. The third dataset is LIMO (Eckmann et al., 2022), a framework for molecular generation aimed at optimizing specific properties by operating in a 1024-dimensional latent space. Further details of these datasets are provided in the Appendix E. All experiments on real-world tasks are minimum optimization.

**The Setting of DSEBO.** DSEBO requires specifying the subspace dimension search range $[d_l, d_h]$, and a hyper-parameter $\beta$. For any high-dimensional problems, the recommended initial subspace dimension is $d_l = 5$, and the upper boundary of the search range is $d_h = \min(D, 100)$, which is the setting for all experiments. Given that the performance of the Bayesian optimizer declines sharply for dimensions above 100, the search range is capped at 100. The hyper-parameter $\beta$ controls the scale of the dimension expansion, and we set $\beta = 12.0$. The shared embedding matrix $S \in \mathbb{R}^{D \times d_h}$ is initialized with elements independently sampled from the Gaussian distribution $\mathcal{N}(0, \sqrt{1/d_h})$.

## 6.1 PERFORMANCE OF HIGH-DIMENSIONAL OPTIMIZATION

In this section, DSEBO is compared against high-dimensional optimization algorithms under limited resources. The comparable methods include REMBO (Wang et al., 2016), SIRBO (Zhang et al., 2019), HesBO (Nayebi et al., 2019), VAEBO (Gómez-Bombarelli et al., 2018), and ALEBO (Letham

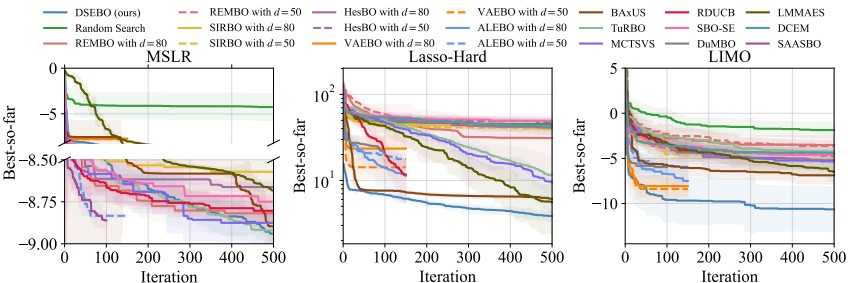

Figure 4: Results on real-world datasets compared with different high-dimensional optimization algorithms. All algorithms are independently repeated 10 times.

et al., 2020), all of which are based on subspace embedding techniques and REMBO serves as an ablated version of DSEBO without the dynamic dimension expanding strategy. We also compare with BAxUS (Papenmeier et al., 2022), which dynamically selects the next subspace dimension. For methods that do not rely on subspace embedding, we conduct TuRBO (Eriksson et al., 2019), SAASBO (Eriksson & Jankowiak, 2021), DuMBO (Bardou et al., 2024), MCTS-VS (Song et al., 2022), RDUCB (Ziomek & Bou-Ammar, 2023), SBO-SE (Xu et al., 2025), LMMAES (Loshchilov et al., 2019), DCEM (Amos & Yarats, 2020) and random search. For further details, refer to Appendix F. All algorithms are tested on high-dimensional optimization tasks with unknown effective dimensions, with hyper-parameters $d = 30, 50$ for synthetic functions and $d = 50, 80$ for real-world tasks. All algorithms are allocated 500 evaluation budget for optimization unless optimization either exceeds 8 hours of runtime (such as SIRBO, ALEBO, VAEBO, etc.) or encounters out-of-memory errors with 16 GB of RAM (such as SAASBO). All algorithms are independently repeated 10 times.

The best-so-far solutions found by the algorithms on partial synthetic functions and real-world datasets are shown in Figure 3 and Figure 4. The horizontal axis represents the number of iterations, with each iteration consuming 1 budget unit. The vertical axis records the best-so-far function value. The remaining results of synthetic functions with $D = 1000$, all results of synthetic functions with $D = 10000$, and detailed results are shown in Appendix G. To further verify DSEBO's capability in handling high-dimensional optimization tasks, we also conduct experiments by adjusting the shifting scalar $c$ of the synthetic functions, with results shown in the Appendix G.

The experimental results show (1) **Superiority.** Compared with other high-dimensional BO algorithms, DSEBO can find optimal solutions across almost all tasks, including real-world tasks and high-dimensional synthetic functions of different magnitudes. (2) **Efficiency.** In high-dimensional tasks with unknown effective dimensions, DSEBO dynamically expands the subspace dimension to find the optimal solution within limited resources. This allows DSEBO to achieve the optimal solution with minimal cost, whereas subspace-based methods require multiple complete optimization processes across different subspace dimensions (Wang et al., 2016) and non-subspace-based methods require optimization in the original high-dimensional space, both consuming significant computational resources. (3) **Continuous Improvement.** Due to its ability to dynamically expand subspace dimensions, the performance of DSEBO continues to improve when other algorithms have converged. When the optimization process converges in the current subspace, DSEBO can expand to a larger subspace for further optimization, ensuring continuous progress. (4) **Adaptability.** DSEBO starts from a lower initial subspace, which makes its initial evaluation values different (such as having a good initial value on synthetic functions while getting poor initial performance on real-world datasets), but DSEBO adapts to the specific tasks and continues to converge towards the optimal solution. (5) **Necessity.** The experimental results show that some high-dimensional embedding BO algorithms (such as REMBO) have obvious performance differences on different subspaces. This illustrates the necessity of designing a dynamic subspace dimension expanding strategy.

## 6.2 PERFORMANCE OF DYNAMIC DIMENSION EXPANDING STRATEGY

To further evaluate the performance of the dynamic dimension expanding strategy, we compare a series of MAB strategies, including Extreme Bandits (Extreme), Classic UCB (C-UCB) (Auer et al., 2002), $\epsilon$-Greedy (Langford & Zhang, 2007), Softmax strategy (Sutton & Barto, 2018), Successive Halving (S-Halving) (Karnin et al., 2013), UCB-E (Audibert et al., 2010), Thompson Sampling (TS) (Jin et al., 2022), Expectation strategy (Expectation), and Random strategy. Each MAB strategy selects different

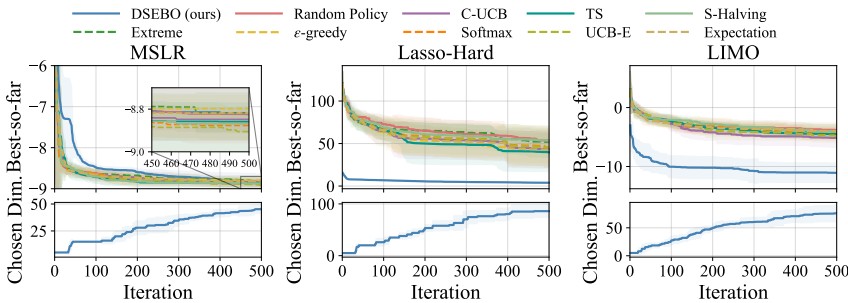

Figure 5: Results on real-world tasks compared with the MAB strategies, and the subspace dimensions selected by DSEBO during the optimization process. All experiments are repeated 10 times.

subspaces for optimization, with dimensions drawn from $\{10, 20, 30, 50, 70, 90, 100\}$. A detailed description of these algorithms is provided in the Appendix F. All strategies are allocated a budget of 500 evaluations, each subspace is initialized by one point, and the experiments are independently repeated 10 times. The experimental results on real-world tasks are presented in Figure 5, showcasing the best-so-far solution found (top) and the subspace dimension expansions (bottom). Results for synthetic functions are provided in the Appendix G.

Experimental results show that: (1) **Superiority and Tailor-Made.** DSEBO designs a dynamic dimension expanding strategy for high-dimensional BO problems with unknown effective dimensions. Compared with the general MAB strategies, it can converge to better performance on various tasks. (2) **Effectiveness of Shared Embedding.** Through experimental results on synthetic functions, it can be found that on tasks with particularly high dimensions, random strategy has the worst results, followed by strategies that explore larger subspace dimensions. Unlike DSEBO, other strategies cannot share data to explore different subspace dimensions. Therefore, exploration in different dimensions brings a huge overhead and reduces the convergence efficiency. (3) **Reasonable Expansions.** The results for four synthetic functions show that when the space dimension is below the effective dimension $d_e$, DSEBO rapidly increases it. However, once the effective dimension is exceeded, the growth rate slows down, reflecting the rationality of DSEBO in expanding subspace dimensions.

### 6.3 Hyper-Parameter Analysis

We conduct hyper-parameter analysis on $\beta$ and the upper boundary of search range $d_h$ to verify the robustness of DSEBO under different hyper-parameter settings. The detailed results and analysis are shown in Appendix H. The hyper-parameter analysis verifies that the chosen hyper-parameter values are reasonable, and shows the robustness of the DSEBO across different settings.

## 7 Conclusion and Discussion

**Conclusion.** This paper introduces the DSEBO method for automated random embedding in high-dimensional Bayesian optimization with unknown effective dimensions. We propose the dynamic shared embedding BO, which dynamically expands the subspace dimension during optimization. Utilizing a shared embedding matrix, one subspace can share the initial dataset from a lower-dimensional subspace. The dynamic dimension expanding strategy determines the dimension of the next subspace, thereby achieving better optimization performance within limited resources. The theoretical analysis establishes a regret bound for DSEBO, proofing its superior ability to balance approximation and optimization errors compared with GPUCB.

**Discussion.** Currently, due to the effective dimension being unknown during the optimization process, DSEBO is unable to halt dimension expansion near the effective dimension, which may lead to unnecessary increases in the subspace dimensionality. In the future, we will explore more adaptive strategies to effectively identify the effective dimension and enable dynamic subspace adjustment (not just dimension expansion) and further integrate the dynamic dimension expanding strategy into other high-dimensional optimization methods, such as evolutionary algorithms, to improve optimization efficiency. Furthermore, we will explore the integration of DSEBO with multiple random embeddings (Cartis et al., 2023) or learned embedding from data (Garnett et al., 2014) approaches to better address high-dimensional optimization tasks with effective dimension.

## 8 ETHICS AND REPRODUCIBILITY STATEMENTS

**Ethics.** This work does not include any human subjects, personal data, or sensitive information. All testing datasets utilized are publicly accessible, and no proprietary or confidential information has been employed.

**Reproducibility.** Experimental settings are described in Section 6 with further details of the methods included in Appendix B and Appendix C. The datasets utilized in this paper are all publicly available and open-source. The link to our anonymous code repository is `https://anonymous.4open. science/r/DSEBO-7532`. No LLMs were used in conducting the research or writing this paper.

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

# A    DETAILED PRELIMINARIES

This section provides additional details on Bayesian optimization, the optimal $\epsilon$-effective dimension, and random embedding to further clarify the preliminaries, necessary background and notation.

## A.1    BAYESIAN OPTIMIZATION

Bayesian optimization (BO) (Srinivas et al., 2010; Shahriari et al., 2016; Garnett, 2023) is a well-known derivative-free optimization method that strategically utilizes prior knowledge to guide the sampling process. First, BO constructs a surrogate model, typically a Gaussian process (GP), based on samples of the objective function. Next, BO calculates the acquisition function based on this model to guide subsequent sampling process and the trade-off between "exploration" and "exploitation" in the search space.

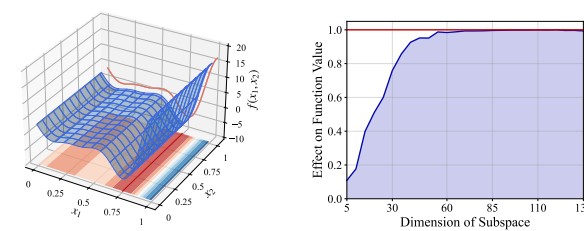

(a) Function with effective dimension

(b) $\epsilon$-effective dimension for MSLR

Figure 6: An illustration for a synthetic function with effective dimension and the test for the existence of $\epsilon$-effective dimension in real dataset MSLR Zhang et al. (2023).

Based on the observed dataset $\mathcal{D}$, BO estimates the posterior distribution of the objective function $P(f|\mathcal{D}) \propto P(\mathcal{D}|f)P(f)$, where $P(f)$ denotes the prior distribution of the objective function and $P(\mathcal{D}|f)$ is the likelihood. This posterior distribution incorporates information about the objective function and is used to inform subsequent modeling and optimization. Once the posterior distribution has been obtained, BO employs an acquisition function to determine the next sampling point. A common choice is UCB (Srinivas et al., 2010), defined as $\alpha_{\mathrm{UCB}}(\boldsymbol{x}) = \mu_t(\boldsymbol{x}) + \sqrt{\kappa_{t+1}}\sigma_t(\boldsymbol{x})$, where $\mu_t(\boldsymbol{x})$ estimates the objective function, associated with "exploitation", while $\sigma_t(\boldsymbol{x})$ represents the uncertainty of the objective function, associated with "exploration". The hyper-parameter $\kappa$ in UCB balances "exploration" and "exploitation" for efficient optimization.

## A.2    RANDOM EMBEDDING

If the high-dimensional objective function has effective dimension, the subspace embedding technique can effectively achieve dimensionality reduction. For example, Figure 6(a) shows a 2-dimensional synthetic function $f(x_1, x_2)$ with a 1-dimensional effective dimension. In such cases, only a few dimensions significantly influence the objective function and are prioritized during optimization.

However, it is challenging to satisfy the effective dimension assumption in real tasks. In this case, Qian et al. (2016) propose the concept of optimal $\epsilon$-effective dimension to relax the assumption, which is defined as Definition 3.1.

Empirical evidence shows that many high-dimensional tasks (Sun et al., 2022) and datasets (Zhang et al., 2023) follow the $\epsilon$-effective dimension. A typical example is the MSLR task (as Figure 6(b)), where 50-dimensional subspace fully captures the variation in the objective function, and other dimensions have a low impact, i.e., the $\epsilon$-effective dimension of the MSLR dataset does not exceed 50.

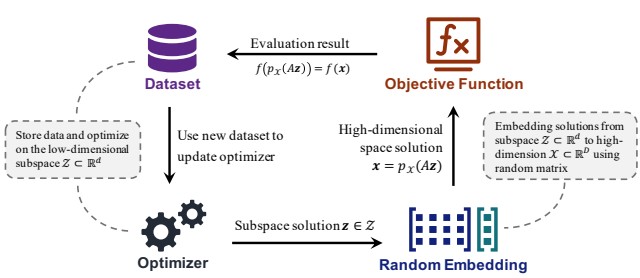

Figure 7: The process of BO with random embedding.

Random embedding (RE) (Wang et al., 2016; Nayebi et al., 2019) is a widely used subspace embedding method as shown in Figure 7. For a high-dimensional optimization problem $\boldsymbol{x}^* = \mathrm{argmin}_{\boldsymbol{x} \in \mathcal{X} \subset \mathbb{R}^D} f(\boldsymbol{x})$, a random matrix $A \in \mathbb{R}^{D \times d}$ with elements sampled from a

Gaussian distribution $\mathcal{N}(0, \sigma^2)$ embeds the $d$-dimensional subspace $\mathcal{Z} \subset \mathbb{R}^d$ into the $D$-dimensional search space $\mathcal{X} \subset \mathbb{R}^D$. The optimizer only needs to optimize $z \in \mathcal{Z}$ in the low-dimensional subspace, and then embed $z$ into $\mathcal{X}$ through the matrix $A$ to get the solution $x = p_\mathcal{X}(Az)$. Here, $p_\mathcal{X}(Az) = \arg\min_{x \in \mathcal{X}} \|x - Az\|_2$ is to project illegal points (i.e., $Az \notin \mathcal{X}$) back to the $\mathcal{X}$ region. Subsequently, the objective function $f$ is evaluated in the solution $x$, and the sample point $(z, f(p_\mathcal{X}(Az)))$ is added to the subspace dataset. The optimizer uses the updated dataset to complete the next optimization iteration.

## B  PSEUDO-CODE OF DSEBO

The pseudo-code of DSEBO is shown in Algorithm 2. The algorithm aims to optimize a high-dimensional objective function $f(\cdot)$ with unknown effective dimensions on a $D$-dimensional space within a limited number of evaluations to find its minimum value. DSEBO requires $budget$ (i.e., total evaluations), the subspace dimension search range $[d_l, d_h]$, and a hyper-parameter $\beta$. For a high-dimensional problem, one can search from the dimension 5, i.e., $d_l = 5$, and the recommended value of the upper boundary of the search range $d_h$ is $\min(D, 100)$. When the dimension exceeds 100, the performance of the Bayesian optimizer will drop sharply, so the search range of the subspace will not exceed 100. The hyper-parameter $\beta$ controls the scale of the dimension expansion.

Before optimization, the algorithm will initialize the shared embedding matrix and set some necessary parameters, as shown in lines 1–2, where $T$ controls the judgment of whether the solution converges, indirectly controlling the number of dimension expansion iterations. During the optimization phase, if a subspace has not been initialized, the algorithm will initialize its dataset, optimizer, current optimal solution $b$, and random embedding matrix in sequence, as shown in lines 4–9. Then, the

---

**Algorithm 2** DSEBO Algorithm

**Input:** $D$-dimensional objective function $f(\cdot)$, $budget$, $d_l = 5$, $d_h = \min(D, 100)$, hyper-parameter $\beta$.

1: Initialize the shared embedding matrix $S_{D \times d_h}$, whose elements are sampled from $\mathcal{N}(0, \sqrt{1/d_h})$
2: $t = 1, d^{(1)} = d_l, T = \lfloor \frac{budget}{2\beta} \rfloor$
3: **while** $t <= budget$ **do**
4:   **if** $t = 1$ or $d^{(t)} \neq d^{(t-1)}$ **then**
5:     $\mathcal{D}^{(t)} \leftarrow$ Initialize dataset using shared embedding
6:     Using $\mathcal{D}^{(t)}$ update the optimizer $B_{d^{(t)}}$
7:     $b \leftarrow \min\{y | (z, y) \in \mathcal{D}^{(t)}\}$
8:     $A_{d^{(t)}} \leftarrow$ The first $d^{(t)}$ rows of matrix $S_{D \times d_h}$
9:   **end if**
10:   $z^{(t)} \leftarrow$ Next solution given by the optimizer $B_{d^{(t)}}$
11:   $x^{(t)} \leftarrow$ Random Embedding $z^{(t)}$, i.e., $p_\mathcal{X}(A_{d^{(t)}} z^{(t)})$
12:   $f(x^{(t)}) \leftarrow$ Query the objective function
13:   $\mathcal{D}^{(t+1)} \leftarrow \mathcal{D}^{(t)} \cup \{(z^{(t)}, f(x^{(t)}))\}$
14:   Using $\mathcal{D}^{(t+1)}$ update the optimizer $B_{d^{(t)}}$
15:   **if** $|f(x^{(t)}) - b| > 0.5$ **then**
16:     $b \leftarrow f(x^{(t)})$
17:   **end if**
18:   **if** $b$ has no change in $T$ iterations **then**
19:     $d^{(t+1)} \leftarrow$ Update dimension using dynamic dimension expanding strategy
20:     $T \leftarrow \lfloor (1 + \frac{d^{(t)} - d_l}{d_h - d_l}) \frac{budget}{2\beta} \rfloor$
21:   **else**
22:     $d^{(t+1)} \leftarrow d^{(t)}$
23:   **end if**
24:   $t \leftarrow t + 1$
25: **end while**
26: $z^* \leftarrow \arg\max_z \{y \mid (z, y) \in \mathcal{D}^{(t)}\}$

**Output:** The best solution $x^* = A^{(t)} z^*$.

---

---

**Algorithm 3** Dataset Initialization

---

**Input:** Previous dimension $d^{(t-1)}$, current dimension $d^{(t)}$, dataset $\mathcal{D}^{(t-1)}$ of previous space.
1: **if** $t = 1$ **then**
2:    $\mathcal{D}^{(1)} \leftarrow$ Add a random $d^{(1)}$-dimensional solution
3: **else**
4:    $\mathcal{D}^{(t)} \leftarrow \varnothing$
5:    **for all** $(\boldsymbol{z}, y) \in \mathcal{D}^{(t-1)}$ **do**
6:       $\boldsymbol{z}' = (\boldsymbol{z}^\top, \boldsymbol{0}_{d^{(t)}-d^{(t-1)}})^\top$
7:       $\mathcal{D}^{(t)} \leftarrow \mathcal{D}^{(t)} \cup \{(\boldsymbol{z}', y)\}$
8:    **end for**
9: **end if**
**Output:** Dataset $\mathcal{D}^{(t)}$ of current space.

---

following step is Bayesian optimization based on stochastic embedding, including steps such as the optimizer giving the next solution, embedding the solution into the high-dimensional search space, querying the objective function, updating the dataset and the optimizer, as shown in lines 10–14. The updating of the optimal value $b$ achievable by the current algorithm occurs in lines 15–17, with the error constrained within the range of $\alpha$. If the solution has converged in the current subspace, i.e., $b$ has not changed in $T$ iterations, expand to the new subspace and update $T$ (the convergence time $T$ of the higher-dimensional subspace solution is usually larger, so we update it), as shown in lines 18–20. Repeat the above steps until the $budget$ is exhausted, and finally embed the subspace into the search space to obtain the optimal value found by the algorithm.

## C   Pseudo-code of Dataset Initialization Algorithm

In high-dimensional optimization problems, when the optimization process enters a new subspace, it is necessary to initialize the dataset in the new dimensional space. To fully utilize the data from the existing lower-dimensional subspace, this algorithm generates the new dataset by extending the existing data into the higher-dimensional space. As shown in Algorithm 3, in the first iteration ($t = 1$), the dataset starts empty and is initialized with a random point in the current dimensional space. For subsequent dimension expansions, the dataset $\mathcal{D}^{(t)}$ for the current dimension is generated by expanding the previous dataset $\mathcal{D}^{(t-1)}$. This expansion is achieved by appending zeros to each solution vector $\boldsymbol{z}$, thereby extending it into the new subspace while preserving the original evaluation value $y$. This method ensures that valuable data from lower dimensions are retained and utilized in the higher-dimensional optimization process.

## D   Proof of Section 5

### D.1   Proof of Theorem 5.1

*Proof.* By Theorem 2 of Srinivas et al. (2010), the regret bound is $O^*\left(\sqrt{T(\log T)^{d+1}}\right)$ in the absence of approximation error. To complete the analysis, we need to account for the impact of the approximation error.

From Lemma 1 of Qian et al. (2016), the approximation error is given by $2\epsilon_d$. Combining these results, the total regret bound becomes the sum of the regret without approximation error and the approximation error term. This completes the proof.     $\square$

### D.2   Proof of Theorem 5.2

*Proof.* Let $\sigma^{(t)}(\cdot)$ be the standard deviation function in step $t$, $r^{(t)}$ be the single-step regret without considering the approximation error. Lemma 5.2 of Srinivas et al. (2010) reveals that $(r^{(t)})^2 =$

$4\beta^{(t)}(\sigma^{(t-1)})^2(\boldsymbol{x}^{(t-1)})$. Therefore,

$$
\begin{aligned}
\sum_{t=1}^{T}(r^{(t)})^2 \\
= \sum_{t=1}^{T} 4\beta^{(t)}(\sigma^{(t-1)})^2 \\
\overset{(a)}{\leq} \sum_{t=1}^{T} 4\beta^{(T)}(\sigma^{(t-1)})^2 \\
= 4\beta^{(T)} \sum_{t=1}^{T}(\sigma^{(t-1)})^2(\boldsymbol{x}^{(t-1)}) \\
= 4\beta^{(T)} \sum_{t=1}^{T} \sigma^2 \left( \sigma^{-2}(\sigma^{(t-1)})^2(\boldsymbol{x}^{(t-1)}) \right) \\
\leq 4\beta^{(T)} \sum_{t=1}^{T} \sigma^2 C_2 \log\left(1 + \sigma^{-2}(\sigma^{(t-1)})^2(\boldsymbol{x}^{(t-1)})\right) \\
\overset{(b)}{\leq} C_1 \beta^{(T)} \sum_{h=1}^{H} \gamma_{T_h},
\end{aligned}
$$

where (a) is because $\beta^{(t)}$ is nondecreasing, (b) comes from Lemma 5.3 of Srinivas et al. (2010).

According to Theorem 5 of Srinivas et al. (2010), we have $\gamma_{T_h} = O(\log T_h)^{d_h+1}$. Plugging this into the above inequality, and use Cauchy-Schwarz inequality as Lemma 5.4 in Srinivas et al. (2010), we obtain the final result. $\qquad\square$

## E  REAL-WORLD DATASETS

In this section, we provide a detailed introduction to the real-world datasets used in our experiments.

**MSLR** (Qin et al., 2010): The first real-world dataset is the Microsoft Learning to Rank (MSLR) dataset (Qin et al., 2010), specifically the MSLR-WEB-10K version, which includes over 10000 queries, each with 136 features per website page. Each website page has a relevance judgment, ranging from 0 (irrelevant) to 4 (completely relevant). When dealing the dataset, we normalize the dataset and then use neural networks to model the dataset as the objective function. Given the complexity of using the dataset as an oracle, we assess preferences through the neural network's predictions instead. The architecture of this neural network is designed with three hidden layers, each containing 128, 64, and 32 neurons, and utilizes the Sigmoid function as the activation mechanism.

**Lasso-Hard** (Sehic et al., 2022): The second dataset is Lasso-Hard from LassoBench (Sehic et al., 2022), a 1000-dimensional optimization task designed to identify sparse regression coefficients that minimize Lasso regression loss.

**LIMO** (Eckmann et al., 2022): The third dataset is LIMO (Eckmann et al., 2022), a framework for molecular generation aimed at optimizing specific properties by operating in a 1024-dimensional latent space learned through a variational autoencoder. Specifically, the LIMO dataset is a drug-like molecule design task, where the objective function is designed to balance the octanol-water partition coefficient, accessibility, and the presence of large rings.

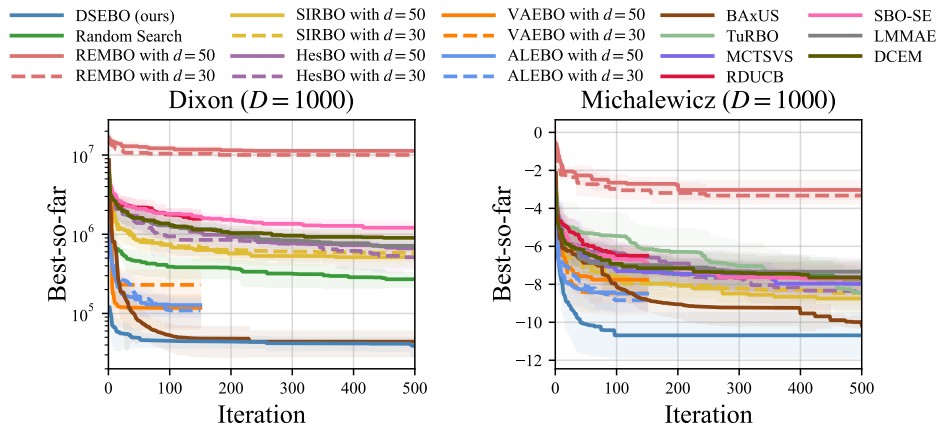

Figure 8: Results on remaining 1000-dimensional synthetic functions compared with various high-dimensional optimization algorithms. All algorithms are independently repeated 10 times.

## F    BASELINES IMPLEMENTATION DETAILS

### F.1    HIGH-DIMENSIONAL OPTIMIZATION METHODS

**REMBO** (Wang et al., 2016): REMBO, the first random embedding approach, repeats using the BoTorch framework in the experiments. We adhere to the same hyper-parameter specifications detailed in (Wang et al., 2016).

**HesBO** (Nayebi et al., 2019): HesBO takes a novel approach to avoid embedding at the boundary by altering the generation process of the embedding matrix. For our implementation, we have utilized the version made available by the author at the GitHub repository: `https://github.com/aminnayebi/HeSBO`.

**SIRBO** (Zhang et al., 2019): Unlike traditional random embedding techniques, SIRBO computes the embedding matrix using the sliced inverse regression (SIR) method. We use the author's implementation from `https://github.com/cjfcsjt/SILBO/blob/master/SIR_BO.py`.

**VAEBO** (Gómez-Bombarelli et al., 2018): VAE-BO employs a variational auto-encoder (VAE) to discern the embedding relationship between high-dimensional spaces and their lower-dimensional counterparts. And we also utilize the code made available by the author: `https://github.com/lamda-bbo/MCTS-VS/blob/master/baseline/vae_bo.py`, adjusting the learning rate to 0.001 and updating the VAE model every 20 iterations.

**ALEBO** (Letham et al., 2020): ALEBO refines the acquisition function within constraints to become the state-of-the-art (SOTA) method for random embedding. We use the author's implementation from `https://github.com/facebookresearch/alebo`.

**BAxUS** (Papenmeier et al., 2022): BAxUS is an embedding-based method designed to optimize high-dimensional black-box functions by using nested random subspaces and a unique dimensionality growth strategy. We use the author's code from `https://github.com/LeoIV/BAxUS`.

**MCTS-VS** (Song et al., 2022): MCTS-VS employs Monte Carlo tree search to iteratively select and optimize a subset of variables within a low-dimensional subspace. The implementation from `https://github.com/lamda-bbo/MCTS-VS` is used.

**TuRBO** (Eriksson et al., 2019): TuRBO is an efficient method for handling high-dimensional optimization by dividing the space into smaller regions for local optimization. We use the code from `https://github.com/uber-research/TuRBO`.

**SAASBO** (Eriksson & Jankowiak, 2021): SAASBO focuses on optimizing only a few important dimensions, thus reducing computational complexity in high-dimensional spaces. We implement the method using the author's code from `https://github.com/martinjankowiak/saasbo`.

**DuMBO** Bardou et al. (2024): DuMBO employs decentralized message-passing and a refined acquisition function to relax additive structure constraints in high-dimensional Bayesian Optimization, achieving asymptotic optimality on functions with complex decompositions. The code for DuMBO can be found at `https://github.com/abardou/dumbo`.

**RDUCB** (Ziomek & Bou-Ammar, 2023): RDUCB uses random tree decompositions to construct additive GP models with cycle-free pairwise dimensional interactions, effectively addressing the challenge of being misled by local data in high-dimensional optimization. The code for RDUCB is available at `https://github.com/huawei-noah/HEBO`.

**SBO-SE** (Xu et al., 2025): SBO-SE employs a robust strategy to initialize the length-scale of the GP kernel, avoiding the vanishing gradient problem during Gaussian process training in high-dimensional Bayesian optimization. The code we use is implemented based on the BoTorch library.

**LMMAES** Loshchilov et al. (2019): LMMAES reduces the time and space complexity of traditional matrix adaptation evolution strategy by approximating the covariance structure using a small set of evolution paths. The code for LMMAES is available as part of the pypop library at `https://github.com/Evolutionary-Intelligence/pypop`.

**DCEM** Amos & Yarats (2020): DCEM is a differentiable variant of the cross-entropy method that enables gradient-based end-to-end learning by employing a smooth top-k operation, addressing the challenges of backpropagating through non-differentiable or discrete optimization steps. The implementation of DCEM can be found in the pypop library at `https://github.com/Evolutionary-Intelligence/pypop`.

### F.2 MULTI-ARMED BANDIT STRATEGY

$\epsilon$-**greedy** (Langford & Zhang, 2007): This method strikes a simple balance between exploration and exploitation by selecting the best known action most of the time while occasionally choosing randomly with a small probability $\epsilon$. This ensures that the algorithm does not rely solely on the existing knowledge and periodically explores other options, potentially discovering more optimal strategies. In our experiments, we set $\epsilon = 0.5$.

**C-UCB** (Auer et al., 2002): Classic Upper Confidence Bound (C-UCB) is the foundational algorithm in the UCB family, which selects actions based on a trade-off between their past rewards and a confidence interval, ensuring a balance between exploiting known rewards and exploring less certain options. In our experiments, the $\kappa$ is set to $\sqrt{2\log(T)/n_i}$ as default, where $T$ is the total number of iterations and $n_i$ is the number of $i$-th subspace optimized time.

**UCB-E** (Audibert et al., 2010): The UCB-E algorithm enhances the traditional Upper Confidence Bound approach by introducing a controllable exploration factor, $c$. This factor allows for fine-tuning the exploration level, independent of each arm's estimated uncertainty, making it especially useful in scenarios where the standard uncertainty model might not adequately represent the exploratory needs. In our experiments, we set $c = 0.5$.

**TS** (Jin et al., 2022): Thompson Sampling (TS) is a probabilistic approach that selects arms based on samples drawn from their estimated reward distributions, effectively balancing exploration and exploitation by adapting to the uncertainty in the estimate of each arm's reward.

**Softmax** (Sutton & Barto, 2018): In the Softmax method, the selection of arms is controlled by a probability distribution weighted by the estimated value of each arm, which is adjusted according to the temperature parameter $\tau$. Lower $\tau$ favors the best arms, while higher $\tau$ increases exploration of unknown arms. In our experiments, we set $\tau = 1.0$.

**S-Halving** (Karnin et al., 2013): Successive Halving is a method of saving budgets by first allocating equal resources to all candidates and then phasing out underperforming candidates, focusing resources on the most promising bandits as the process iterates.

**Extreme**: The extreme strategy is designed to achieve exploitation by aggressively focusing on the arms that appear to be optimal, often used in environments where the cost of exploration is high or when quick decisions are critical.

**Expectation**: The expectation strategy selects arms based on the expected value of the reward, typically favoring arms with higher average rewards.

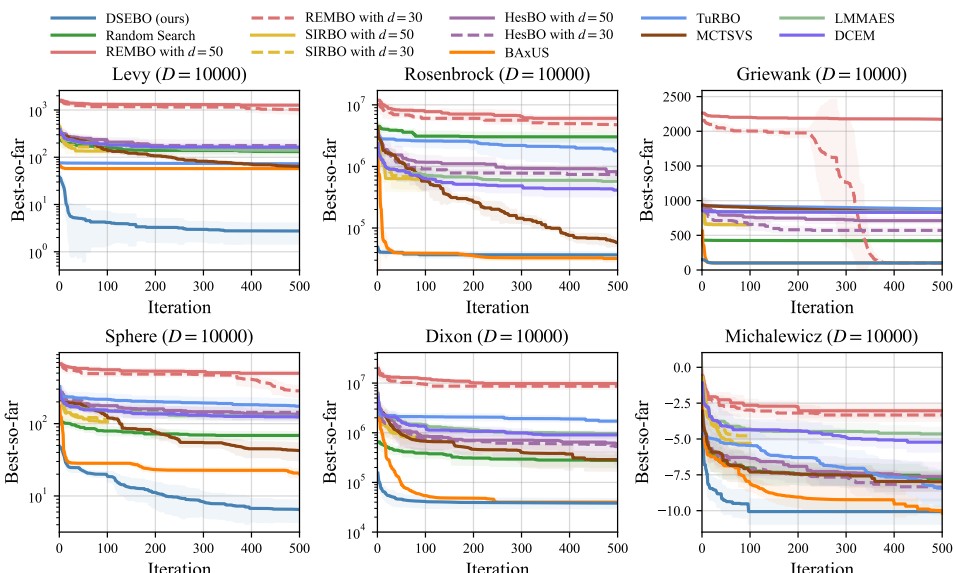

Figure 9: Results on 10000-dimensional synthetic functions compared with various high-dimensional optimization algorithms. All algorithms are independently repeated 10 times.

**Random**: The random strategy selects actions randomly, ensuring equal exploration of all available options. It is useful for baseline comparisons or when no prior data exists. Although simple, it can occasionally reveal overlooked possibilities in complex scenarios.

## G  DETAILED RESULTS

The remaining experimental results of the synthetic functions with $D = 1000$ mentioned in the Section 6 are shown in Figure 8. The experimental results of the synthetic functions with $D = 10000$ are shown in Figure 9. When dealing with high-dimensional tasks with $D = 10000$, since SIRBO cannot complete 500 iterations within 8 hours, only the results of the first 100 iterations are plotted. Experimental results show that on 6 synthetic functions, DSEBO shows the fastest convergence speed and can find solutions with high performance.

Additionally, the comparison results of synthetic functions between DSEBO and a series of MAB strategies are presented in Figure 10. As evidenced by the figures, DSEBO verifies significant advantages over other arm selection strategies by dynamically expanding subspace dimensions, efficiently sharing data across dimensions, and achieving faster convergence and superior performance in high-dimensional optimization tasks.

Table 1, Table 2, Table 3, Table 4 and Table 5 record the final mean convergence value of various algorithms under each experimental environment, the optimal solution that can be found, and the mean operation time. The results show that DSEBO can provide a better dynamic dimension expanding strategy and show good optimization performance in all synthetic functions and real-world tasks, which reflects in stable convergence performance and the ability to explore excellent solutions.

To further evaluate the capability of DSEBO, we conduct experiments on the Sphere function by shifting the optimal solution's location through varying $c$, as shown in Figure 11, which verifies that DSEBO consistently maintains its superior performance across different functions.

## H  HYPER-PARAMETER ANALYSIS

We conduct hyper-parameter analysis on $\beta$ and the upper boundary of search range $d_h$ on two synthetic functions to verify the robustness of DSEBO under different hyper-parameter configurations, as shown in Figure 12 and Figure 13.

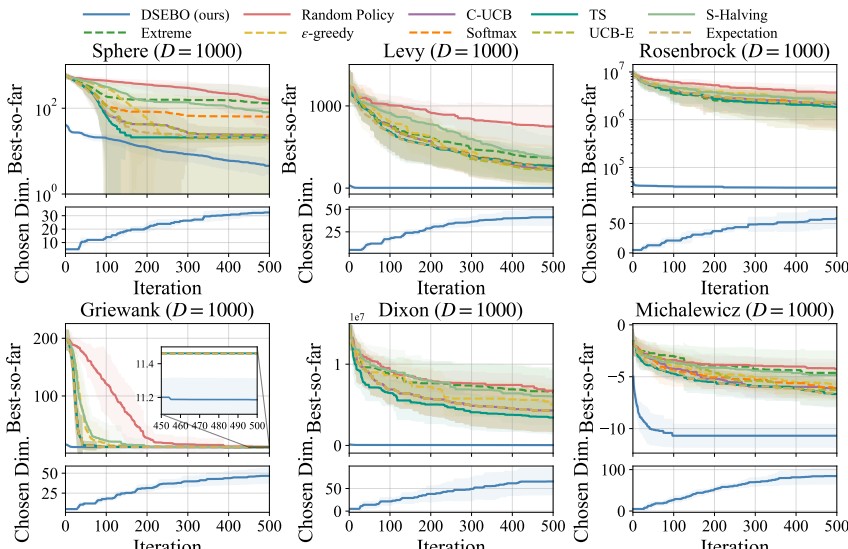

Figure 10: Results on synthetic functions compared with different MAB strategies, and the subspace dimensions selected by DSEBO throughout the optimization process. All algorithms are independently repeated 10 times.

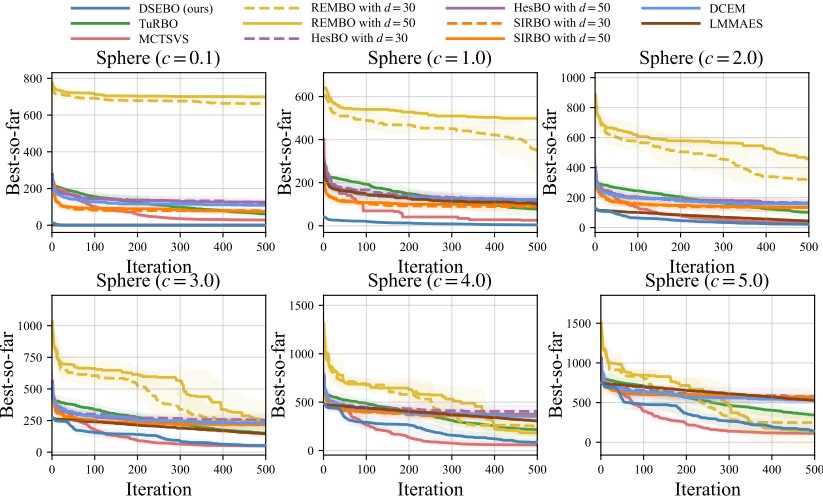

Figure 11: Results on the 1000-dimensional Sphere function with an effective dimension of 30 for varying $c$, compared with various algorithms. All algorithms are independently repeated 10 times.

The experiments of hyper-parameter $\beta$ show that when $\beta$ is too large (e.g., $\beta = 24, 32$), the dimension updates more frequently, but the change in dimensionality is very small. Conversely, when $\beta$ is too small (e.g., $\beta = 1, 4$), the change in dimensionality becomes less intuitive, significantly affecting the convergence of the solution. Nevertheless, DSEBO consistently finds good solutions across different $\beta$ configurations, except for the extreme hyper-parameter values (e.g., $\beta = 1, 4$). From the experiments of the upper boundary of search range $d_h$, it can be observed that setting $d_h = 100$ yields satisfactory optimization performance. Moreover, compared to $d_h = 80$ and $d_h = 120$, $d_h = 100$ results in a more reasonable subspace dimension selection that does not simply opt for higher dimensions indiscriminately.

The hyper-parameter analysis verifies that the chosen hyper-parameter values are reasonable, and shows the robustness of the DSEBO across different settings.

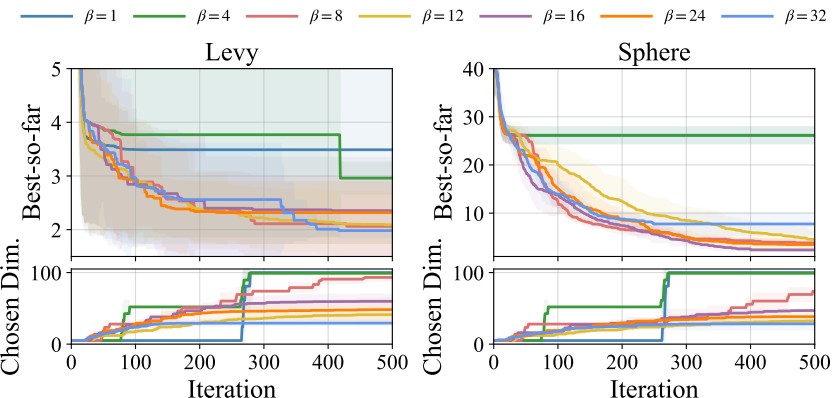

Figure 12: Results of hyper-parameter $\beta$ experiments in the synthetic functions with $D = 1000$, $d_e = 30$. All experiments are repeated 10 times.

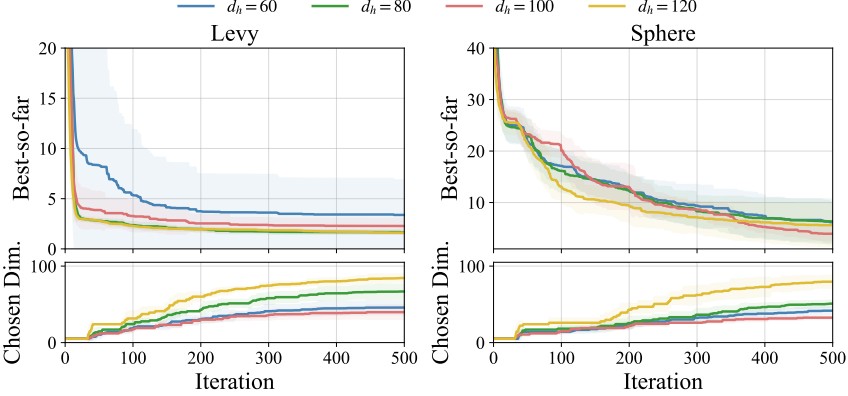

Figure 13: Results of hyper-parameter $d_h$ experiments in the synthetic functions with $D = 1000$, $d_e = 30$. All experiments are repeated 10 times.

Table 1: Detailed results of high-dimensional optimization algorithms on synthetic functions with $D = 1000$.

| Objective Function | Method | Convergence Value | | Best Solution | | Time (s) |
|---|---|---|---|---|---|---|
| | | d=30 | d=50 | d=30 | d=50 | |
| Rosenbrock (D=1000) | SIRBO | 413740.1875±120969.2969 | 459225.1875±96250.1875 | 259375.1562 | 348556.6562 | 54 |
| | HesBO | 633849.8750±200530.8281 | 819959.5000±137244.8906 | 341874.2500 | 681234.4375 | 125 |
| | REMBO | 6223221.1331±641472.7863 | 6183677.2826±809638.4103 | 5236903.2094 | 5851605.9492 | 766 |
| | ALEBO | 121149.1016±46185.1758 | 109807.8828±15043.3242 | 81243.7891 | 92437.4062 | 27665 |
| | VAEBO | 81353.2500±13184.3848 | 89912.2031±11958.9248 | 63902.0126 | 79479.9531 | 21107 |
| | BAxUS | **33332.2138±2957.4265** | | **23831.3776** | | 1241 |
| | MCTSVS | 129055.41±59201.297 | | 70404.7031 | | 57 |
| | TuRBO | 79969.6875±14060.7158 | | 56217.6797 | | 156 |
| | RDUCB | 915660.5000±222155.5938 | | 556649.5465 | | 16653 |
| | SBO-SE | 643878.8125±201391.4688 | | 254839.0312 | | 1334 |
| | DCEM | 527429.5391±83910.8251 | | 390377.3541 | | 7 |
| | LMMAES | 447685.8176±110445.6733 | | 252948.5421 | | 14 |
| | Random Search | 2509598.2157±520699.5662 | | 1677970.2069 | | 4 |
| | DSEBO (ours) | 37699.6758±780.7019 | | 37010.5820 | | 1046 |
| Sphere (D=1000) | SIRBO | 87.9686±15.1527 | 95.3077±7.7289 | 49.4295 | 85.0751 | 90 |
| | HesBO | 120.2445±13.1439 | 114.5023±28.0399 | 99.4486 | 70.6100 | 305 |
| | REMBO | 350.8991±98.4352 | 498.2905±18.5448 | 192.5025 | 477.4872 | 453 |
| | ALEBO | 31.3825±10.4225 | 37.4376±2.5337 | 19.4079 | 35.9000 | 26648 |
| | VAEBO | 37.4775±6.6590 | 37.6364±5.9088 | 25.1240 | 31.1741 | 15163 |
| | BAxUS | 17.8187±2.7321 | | 14.4317 | | 785 |
| | MCTSVS | 36.5522±4.2780 | | 29.4209 | | 51 |
| | TuRBO | 29.7788±11.4849 | | 14.512 | | 137 |
| | RDUCB | 174.8453±12.5177 | | 151.0879 | | 13759 |
| | SBO-SE | 152.2971±9.8699 | | 135.7395 | | 506 |
| | DCEM | 122.1613±15.5641 | | 100.8576 | | 11 |
| | LMMAES | 99.1157±15.992 | | 76.3501 | | 12 |
| | Random Search | 46.3139±37.8249 | | 40.1777 | | 5 |
| | DSEBO (ours) | **3.9387±1.3304** | | **2.4966** | | 339 |
| Levy (D=1000) | SIRBO | 90.0148±16.1821 | 97.3569±11.5434 | 63.7397 | 77.0712 | 95 |
| | HesBO | 149.0245±36.4953 | 146.9163±18.3832 | 117.1409 | 108.5397 | 295 |
| | REMBO | 1046.6322±67.1422 | 1206.4437±76.6944 | 927.1597 | 1087.9023 | 756 |
| | ALEBO | 35.7776±1.6821 | 35.0879±9.3454 | 34.5007 | 24.4679 | 25213 |
| | VAEBO | 18.6543±6.8300 | 19.4487±8.2072 | 11.1254 | 8.6804 | 15323 |
| | BAxUS | 6.7886±0.0155 | | 6.7753 | | 630 |
| | MCTSVS | 32.6922±8.1272 | | 18.0643 | | 118 |
| | TuRBO | 6.4294±0.1092 | | 6.2491 | | 185 |
| | RDUCB | 198.5230±27.5693 | | 146.3279 | | 13496 |
| | SBO-SE | 184.6281±29.2152 | | 110.4856 | | 753 |
| | DCEM | 134.6127±14.4924 | | 110.9705 | | 9 |
| | LMMAES | 98.4986±9.0345 | | 81.0548 | | 11 |
| | Random Search | 113.8001±18.2524 | | 88.0861 | | 5 |
| | DSEBO (ours) | **2.2816±0.7567** | | **1.5805** | | 268 |
| Griewank (D=1000) | SIRBO | 61.7318±0.9419 | 62.0860±0.9048 | 60.1888 | 60.1376 | 92 |
| | HesBO | 64.8294±8.3552 | 73.6908±5.3223 | 44.9517 | 63.0216 | 315 |
| | REMBO | 11.4265±0.8796 | 220.8467±2.4297 | 11.0122 | 215.4218 | 639 |
| | ALEBO | 18.1015±1.3217 | 18.4540±1.2353 | 16.8223 | 17.3414 | 23792 |
| | VAEBO | 14.6719±0.4778 | 13.9477±0.6756 | 14.1418 | 12.7958 | 14685 |
| | BAxUS | 11.2585±0.0474 | | 11.1369 | | 1360 |
| | MCTSVS | 62.4722±1.7076 | | 60.0604 | | 64 |
| | TuRBO | 63.1086±1.5921 | | 61.1658 | | 259 |
| | RDUCB | 164.7312±56.6559 | | 74.8729 | | 15583 |
| | SBO-SE | 88.7530±1.0904 | | 86.7066 | | 843 |
| | DCEM | 80.5171±1.4105 | | 77.9347 | | 10 |
| | LMMAES | 83.7417±2.1408 | | 79.8172 | | 12 |
| | Random Search | 41.1532±0.4551 | | 40.3172 | | 4 |
| | DSEBO (ours) | **11.2488±0.1092** | | **10.9793** | | 276 |
| Dixon (D=1000) | SIRBO | 582309.6250±127018.3516 | 512858.5000±120621.3672 | 399245.0938 | 282305.4062 | 116 |
| | HesBO | 512598.8125±150171.8906 | 662775.3750±159014.9062 | 314195.2812 | 361927.0312 | 148 |
| | REMBO | 10028901.2305±751474.3750 | 11296827.1480±432756.4375 | 8892939.2391 | 10683308.8750 | 531 |
| | ALEBO | 108987.9219±41572.4883 | 127629.6016±37048.0312 | 47649.1914 | 94961.3281 | 23283 |
| | VAEBO | 229556.6719±119026.5547 | 117513.2344±54179.7344 | 111616.2344 | 68923.3125 | 11638 |
| | BAxUS | 44058.0664±15399.4717 | | 26695.2305 | | 953 |
| | MCTSVS | 126604.6875±52375.7891 | | 56397.1484 | | 58 |
| | TuRBO | 738974.3125±436899.8750 | | 111616.2344 | | 218 |
| | RDUCB | 1566930.6250±470125.1562 | | 825860.3044 | | 16561 |
| | SBO-SE | 1208742.0000±248711.9062 | | 864583.7500 | | 657 |
| | DCEM | 900571.3348±214454.3493 | | 495204.757 | | 11 |
| | LMMAES | 682351.8439±98667.3857 | | 515423.2172 | | 13 |
| | Random Search | 270112.6082±53654.5844 | | 184623.3812 | | 11 |
| | DSEBO (ours) | **39076.9609±10246.4609** | | **25189.1094** | | 597 |
| Michalewicz (D=1000) | SIRBO | -8.4669±0.5782 | -8.7477±0.7193 | -9.3439 | -9.8313 | 144 |
| | HesBO | -8.3571±0.9363 | -7.6276±0.9776 | -9.9054 | -9.1226 | 136 |
| | REMBO | -3.3269±0.4194 | -3.0310±0.4671 | -3.9171 | -3.7214 | 784 |
| | ALEBO | -8.8393±0.5495 | -8.4900±0.9351 | -9.7144 | -10.0362 | 26832 |
| | VAEBO | -8.4725±1.0752 | -7.7547±1.2024 | -10.5696 | -10.0460 | 11251 |
| | BAxUS | -10.1528±0.3488 | | -10.4914 | | 1072 |
| | MCTSVS | -7.9805±0.6546 | | -8.9399 | | 34 |
| | TuRBO | -8.4346±1.2460 | | -10.4085 | | 246 |
| | RDUCB | -6.4992±0.5495 | | -7.8331 | | 14549 |
| | SBO-SE | -7.7120±0.6833 | | -8.6905 | | 542 |
| | DCEM | -7.6842±0.8374 | | -9.2082 | | 8 |
| | LMMAES | -7.3345±0.4382 | | -7.9377 | | 14 |
| | Random Search | -7.7538±0.5373 | | -9.0669 | | 16 |
| | DSEBO (ours) | **-10.6887±1.1657** | | **-12.9019** | | 465 |

Table 2: Detailed results of high-dimensional optimization algorithms on real-world tasks.

| Objective Function | Method | Convergence Value | | Best Solution | | Time (s) |
|---|---|---|---|---|---|---|
| | | d=50 | d=80 | d=50 | d=80 | |
| MSLR | SIRBO | -8.6166±0.0063 | -8.5711±0.0094 | -8.6594 | -8.6050 | 71 |
| | HesBO | -8.7637±0.0161 | -8.6621±0.0272 | -8.8077 | -8.8044 | 219 |
| | REMBO | -8.0890±0.0195 | -8.8185±0.0145 | -8.8886 | -8.8850 | 252 |
| | ALEBO | -8.8336±0.0112 | -8.8620±0.0197 | -8.8814 | -8.8962 | 24016 |
| | VAEBO | -7.7777±0.1623 | -7.7285±0.1480 | -8.5001 | -8.2889 | 9766 |
| | BAxUS | -8.8998±0.0964 | | -9.0185 | | 554 |
| | MCTSVS | -8.8755±0.1140 | | -9.0446 | | 49 |
| | TuRBO | -8.9199±0.1023 | | -9.0642 | | 212 |
| | SAASBO | -8.8604±0.0472 | | -8.9474 | | 14326 |
| | RDUCB | -8.8035±0.0944 | | -8.9027 | | 3776 |
| | DuMBO | -8.7808±0.1257 | | -8.9408 | | 6873 |
| | SBO-SE | -8.7492±0.1289 | | -8.9026 | | 307 |
| | DCEM | -8.7281±0.0806 | | -8.8316 | | 5 |
| | LMMAES | -8.6829±0.1933 | | **-9.2051** | | 8 |
| | Random Search | -4.2603±1.4577 | | -7.5904 | | 2 |
| | DSEBO (ours) | **-8.9396±0.0914** | | -9.1933 | | 245 |
| Lasso-Hard | SIRBO | 39.8469±2.7875 | 40.0981±1.9785 | 37.9968 | 37.7477 | 145 |
| | HesBO | 41.2607±11.5219 | 47.6895±8.0687 | 24.9073 | 37.6182 | 317 |
| | REMBO | 31.1405±21.1006 | 41.2012±7.1255 | 6.7579 | 34.3955 | 740 |
| | ALEBO | 11.2730±5.7479 | 17.7599±3.1188 | 8.0080 | 12.9113 | 21846 |
| | VAEBO | 23.4285±9.4164 | 14.2004±3.6496 | 10.9416 | 9.8085 | 13657 |
| | BAxUS | 6.1045±0.5563 | | 5.1150 | | 642 |
| | MCTSVS | 9.4854±2.7239 | | 7.6786 | | 133 |
| | TuRBO | 11.5030±7.3585 | | 4.2641 | | 276 |
| | RDUCB | 11.6843±6.2225 | | 4.4439 | | 14924 |
| | DuMBO | 20.0604±4.6829 | | 12.6193 | | 14700 |
| | SBO-SE | 49.3089±4.2530 | | 42.7787 | | 584 |
| | DCEM | 43.6111±6.5621 | | 32.1892 | | 96 |
| | LMMAES | 5.5371±3.2662 | | **1.8479** | | 112 |
| | Random Search | 45.2683±1.5762 | | 42.5421 | | 29 |
| | DSEBO (ours) | **3.8613±0.4896** | | 3.4248 | | 483 |
| LIMO | SIRBO | -5.2516±0.8961 | -5.1352±0.4046 | -6.8275 | -5.8603 | 126 |
| | HesBO | -5.2117±1.4180 | -4.7430±0.8296 | -7.7866 | -6.2939 | 229 |
| | REMBO | -3.526±0.9881 | -3.6328±0.3971 | -5.0612 | -4.3895 | 164 |
| | ALEBO | -7.5033±1.5741 | -5.9202±0.7087 | -10.4432 | -6.8159 | 25496 |
| | VAEBO | -8.0703±1.3382 | -8.3918±2.0633 | -9.7784 | -11.4423 | 14205 |
| | BAxUS | -6.8716±0.9177 | | -8.065 | | 1562 |
| | MCTSVS | -5.3277±0.7252 | | -6.5733 | | 104 |
| | TuRBO | -4.2479±1.3035 | | -6.9445 | | 291 |
| | RDUCB | -4.0750±0.6278 | | -5.4911 | | 16256 |
| | DuMBO | -5.9619±0.8675 | | -7.0755 | | 21299 |
| | SBO-SE | -4.6146±0.6944 | | -5.8465 | | 544 |
| | DCEM | -4.3841±0.4846 | | -5.3127 | | 114 |
| | LMMAES | -6.5043±1.1578 | | -7.805 | | 126 |
| | Random Search | -1.8629±0.6874 | | -3.0679 | | 12 |
| | DSEBO (ours) | **-10.6613±2.4279** | | **-14.2513** | | 294 |

Table 3: Detailed results of high-dimensional optimization algorithms on synthetic functions with $D = 10000$.

| Objective Function | Method | Convergence Value | | Best Solution | | Time (s) |
|---|---|---|---|---|---|---|
| | | d=30 | d=50 | d=30 | d=50 | |
| Rosenbrock (D=10000) | SIRBO | 654297.3750±251160.9062 | 620616.1250±226600.6094 | 391632.6250 | 429690.8438 | 7255 |
| | HesBO | 740927.6875±95669.8203 | 823125.9375±180349.7031 | 621726.5464 | 558716.9375 | 159 |
| | REMBO | 4762362.9526±1058705.0084 | 6043180.2348±539507.1027 | 3476606.4671 | 5216455.1280 | 795 |
| | BAxUS | **34236.7869±2550.1297** | | **29355.3671** | | 3995 |
| | MCTSVS | 57851.9000±6859.1123 | | 50801.8242 | | 272 |
| | TuRBO | 1788346.6±1056224.8 | | 404563.2812 | | 389 |
| | DCEM | 412348.2053±73896.1685 | | 272307.0427 | | 31 |
| | LMMAES | 572015.3508±114679.9362 | | 357881.9302 | | 54 |
| | Random Search | 3022829.2621±516710.6428 | | 2262149.8444 | | 18 |
| | DSEBO (ours) | 36658.1680±4038.2712 | | 31940.3965 | | 1076 |
| Sphere (D=10000) | SIRBO | 114.9471±17.8656 | 106.1076±12.5741 | 92.1035 | 85.4588 | 7528 |
| | HesBO | 143.2573±26.0684 | 135.1932±21.8210 | 113.4040 | 103.8779 | 230 |
| | REMBO | 282.7653±54.2532 | 497.3164±34.3274 | 198.8264 | 446.1657 | 581 |
| | BAxUS | 20.0685±2.4054 | | 15.9891 | | 3182 |
| | MCTSVS | 42.4199±11.5857 | | 24.8201 | | 383 |
| | TuRBO | 174.1312±39.0554 | | 118.9279 | | 264 |
| | DCEM | 124.469±14.4034 | | 100.0146 | | 30 |
| | LMMAES | 123.6035±14.0609 | | 97.1999 | | 50 |
| | Random Search | 68.6981±8.4385 | | 53.9599 | | 14 |
| | DSEBO (ours) | **6.4338±2.1632** | | **4.6624** | | 343 |
| Levy (D=10000) | SIRBO | 151.7623±8.6825 | 134.7558±19.8493 | 139.4174 | 112.8558 | 7632 |
| | HesBO | 176.8307±52.7794 | 147.1899±18.5760 | 119.6019 | 131.4840 | 155 |
| | REMBO | 1012.7800±231.5087 | 1266.2434±85.5595 | 606.2893 | 1168.6647 | 562 |
| | BAxUS | 57.4391±0.0222 | | 57.4031 | | 3038 |
| | MCTSVS | 64.1901±5.2330 | | 58.3096 | | 451 |
| | TuRBO | 72.8131±0.4172 | | 72.2262 | | 356 |
| | DCEM | 164.0851±22.484 | | 126.7344 | | 41 |
| | LMMAES | 140.5668±13.6518 | | 120.7209 | | 65 |
| | Random Search | 131.8982±18.8193 | | 102.7094 | | 17 |
| | DSEBO (ours) | **2.7522±1.2717** | | **1.8678** | | 290 |
| Griewank (D=10000) | SIRBO | 655.2772±17.9869 | 652.7079±12.9206 | 636.1684 | 643.0330 | 8631 |
| | HesBO | 571.2280±34.4433 | 711.1620±39.2822 | 522.5161 | 672.2633 | 157 |
| | REMBO | **101.1997±0.0365** | 2173.1267±9.0637 | 101.1423 | 2161.6895 | 537 |
| | BAxUS | 101.9942±0.5218 | | 101.4823 | | 3092 |
| | MCTSVS | 835.3915±6.4354 | | 825.446 | | 267 |
| | TuRBO | 882.3385±3.0383 | | 878.386 | | 382 |
| | DCEM | 830.4092±3.2544 | | 825.6568 | | 28 |
| | LMMAES | 840.1342±2.4772 | | 835.4587 | | 57 |
| | Random Search | 422.5575±1.2538 | | 420.7823 | | 16 |
| | DSEBO (ours) | 101.4063±0.0356 | | 101.1153 | | 527 |
| Dixon (D=10000) | SIRBO | 764620.5000±178713.5156 | 711741.6875±220006.9375 | 504689.5000 | 393210.5938 | 6855 |
| | HesBO | 527328.7500±293799.1562 | 593128.7500±238018.3594 | 195464.7500 | 247606.3281 | 195 |
| | REMBO | 8629979.1862±1030159.7500 | 9811402.4797±1358266.3750 | 7375704.5000 | 7227984.3564 | 658 |
| | BAxUS | 40101.4234±7970.3999 | | 34711.2812 | | 3463 |
| | MCTSVS | 288137.3750±118116.6797 | | 138630.5781 | | 250 |
| | TuRBO | 1712952.6250±153219.5156 | | 1534449.3750 | | 294 |
| | DCEM | 906037.6463±127875.9288 | | 757046.0837 | | 33 |
| | LMMAES | 967289.2009±145768.8284 | | 658107.9851 | | 55 |
| | Random Search | 277199.8422±75239.0032 | | 107642.0207 | | 19 |
| | DSEBO (ours) | **38253.0625±8405.7793** | | **23499.7630** | | 366 |
| Michalewicz (D=10000) | SIRBO | -5.2707±0.8001 | -5.4101±0.5353 | -6.2494 | -6.1494 | 5983 |
| | HesBO | -5.3715±0.6174 | -5.4323±0.6975 | -6.4223 | -6.5907 | 803 |
| | REMBO | 0.7975±0.2896 | 1.0713±0.1826 | 0.3354 | 0.8087 | 579 |
| | BAxUS | -10.3466±1.5608 | | -11.8395 | | 3992 |
| | MCTSVS | -10.7175±1.1713 | | -11.5057 | | 342 |
| | TuRBO | -4.2744±0.8549 | | -5.4950 | | 276 |
| | DCEM | -5.2218±0.6693 | | -6.3025 | | 37 |
| | LMMAES | -4.6498±0.4431 | | -5.4561 | | 62 |
| | Random Search | -3.2545±0.4323 | | -3.9067 | | 14 |
| | DSEBO (ours) | **-11.7390±0.6309** | | **-12.7581** | | 391 |

Table 4: Detailed results of MAB strategies on synthetic functions with $D = 1000$.

| Objective Function | Method | Convergence Value | Best Solution | Time (s) |
|---|---|---|---|---|
| Rosenbrock (D=1000) | Extreme | 2160763.7500±1467483.8750 | 571516.5625 | 359 |
| | Random | 3700211.5000±1493292.5000 | 1157037.0000 | 371 |
| | $\epsilon$-greedy | 2273714.5000±1358569.0000 | 52278.8203 | 1193 |
| | C-UCB | 2211636.7500±1624777.0000 | 779282.4375 | 395 |
| | Softmax | 2211636.7500±1624777.0000 | 779282.4375 | 398 |
| | TS | 1856110.0000±834666.1875 | 383319.5625 | 474 |
| | UCB-E | 2248272.2500±1751758.2500 | 779282.4375 | 463 |
| | S-Halving | 2328358.7500±1153945.5000 | 463237.6562 | 401 |
| | Expectation | 2248154.2500±1751778.3750 | 779282.4375 | 555 |
| | DSEBO (ours) | **37699.6758±780.7019** | **37010.5820** | 1046 |
| Sphere (D=1000) | Extreme | 129.6299±181.8460 | 13.1881 | 1223 |
| | Random | 150.9959±121.5409 | 25.4365 | 146 |
| | $\epsilon$-greedy | 19.5962±3.7292 | 13.1885 | 468 |
| | C-UCB | 22.8336±7.7990 | 13.1881 | 1603 |
| | Softmax | 63.3230±127.7900 | 13.1881 | 1348 |
| | TS | 20.9170±4.5001 | 13.1881 | 1791 |
| | UCB-E | 22.8336±7.7990 | 13.1881 | 1555 |
| | S-Halving | 80.7406±113.8652 | 13.1898 | 939 |
| | Expectation | 20.5691±4.5567 | 13.1881 | 1613 |
| | DSEBO (ours) | **3.9387±1.3304** | **2.4966** | 399 |
| Levy (D=1000) | Extreme | 363.8994±380.7985 | 4.2985 | 1013 |
| | Random | 750.0236±276.8358 | 40.4695 | 134 |
| | $\epsilon$-greedy | 217.8715±167.7238 | 2.9233 | 411 |
| | C-UCB | 233.7007±167.9456 | 4.2985 | 1289 |
| | Softmax | 272.7462±251.0918 | 4.2985 | 1393 |
| | TS | 267.9188±218.1308 | 4.2985 | 1288 |
| | UCB-E | 217.7698±150.8256 | 4.2985 | 1268 |
| | S-Halving | 360.4880±277.0266 | 18.9699 | 976 |
| | Expectation | 222.1218±201.5956 | 4.2985 | 1227 |
| | DSEBO (ours) | **2.2816±0.7567** | **1.5805** | 268 |
| Griewank (D=1000) | Extreme | 11.4613±0.3793 | 11.2658 | 1285 |
| | Random | 11.8220±1.5041 | 11.2802 | 95 |
| | $\epsilon$-greedy | 11.5547±0.6718 | 11.2663 | 365 |
| | C-UCB | 11.4613±0.3793 | 11.2658 | 1285 |
| | Softmax | 11.4613±0.3793 | 11.2658 | 1289 |
| | TS | 11.4613±0.3793 | 11.2658 | 1287 |
| | UCB-E | 11.4613±0.3793 | 11.2658 | 1290 |
| | S-Halving | 11.6494±1.0168 | 11.2120 | 1047 |
| | Expectation | 11.4613±0.3793 | 11.2658 | 1301 |
| | DSEBO (ours) | **11.2488±0.1092** | **10.9793** | 276 |
| Dixon (D=1000) | Extreme | 6540823.5000±2834092.7500 | 1504931.8750 | 455 |
| | Random | 6720030.5000±1582640.6250 | 4071802.5000 | 592 |
| | $\epsilon$-greedy | 5106037.5000±1698677.7500 | 2753771.2500 | 362 |
| | C-UCB | 4299205.5000±2713686.0000 | 51601.1992 | 465 |
| | Softmax | 4299205.5000±2713686.0000 | 51601.1992 | 604 |
| | TS | 3428899.5000±2103435.0000 | 33287.6367 | 621 |
| | UCB-E | 4299205.5000±2713686.0000 | 51601.1992 | 716 |
| | S-Halving | 6044710.5000±3704331.5000 | 2606495.7500 | 1124 |
| | Expectation | 4299205.5000±2713686.0000 | 51601.1992 | 603 |
| | DSEBO (ours) | **39076.9609±10246.4609** | **25189.1094** | 597 |
| Michalewicz (D=1000) | Extreme | -4.6286±2.2142 | -8.8314 | 503 |
| | Random | -4.2663±0.8069 | -5.8979 | 92 |
| | $\epsilon$-greedy | -5.7135±1.5460 | -7.9152 | 271 |
| | C-UCB | -6.0991±0.9943 | -7.8918 | 366 |
| | Softmax | -6.0994±0.9661 | -7.3633 | 570 |
| | TS | -6.6686±1.2886 | -8.8314 | 284 |
| | UCB-E | -6.2666±1.0264 | -7.5300 | 204 |
| | S-Halving | -4.8470±1.2241 | -6.6878 | 429 |
| | Expectation | -6.5533±1.3236 | -8.8314 | 344 |
| | DSEBO (ours) | **-10.6887±1.1657** | **-12.9019** | 465 |

Table 5: Detailed results of MAB strategies on real-world tasks.

| Objective Function | Method | Convergence Value | Best Solution | Time (s) |
|---|---|---|---|---|
| MSLR | Extreme | -8.8175±0.0982 | -8.9992 | 548 |
| | Random | -8.8245±0.1437 | -9.0390 | 129 |
| | $\epsilon$-greedy | -8.7972±0.0639 | -8.9243 | 346 |
| | C-UCB | -8.8482±0.0580 | -8.9549 | 494 |
| | Softmax | -8.8766±0.0657 | -9.0177 | 663 |
| | TS | -8.8586±0.0928 | -9.0130 | 744 |
| | UCB-E | -8.9054±0.1266 | -9.1812 | 332 |
| | S-Halving | -8.8690±0.1694 | -9.1396 | 557 |
| | Expectation | -8.8185±0.1308 | -9.1285 | 627 |
| | DSEBO (ours) | **-8.9396±0.0914** | **-9.1933** | 245 |
| Lasso-Hard | Extreme | 52.0638±19.5580 | 7.6819 | 402 |
| | Random | 52.3876±15.1716 | 15.5185 | 342 |
| | $\epsilon$-greedy | 44.1027±17.8642 | 7.4094 | 438 |
| | C-UCB | 43.4808±23.4663 | 7.4081 | 819 |
| | Softmax | 45.7185±25.1736 | 7.4081 | 821 |
| | TS | 39.6904±24.0790 | 6.7548 | 783 |
| | UCB-E | 46.0051±22.9438 | 7.6461 | 382 |
| | S-Halving | 53.6740±14.7570 | 27.8280 | 405 |
| | Expectation | 42.9734±21.6694 | 7.4081 | 795 |
| | DSEBO (ours) | **3.8613±0.4896** | **3.4248** | 483 |
| LIMO | Extreme | -3.9495±1.0746 | -5.7674 | 406 |
| | Random | -3.7930±1.0581 | -5.2494 | 161 |
| | $\epsilon$-greedy | -4.0860±0.8866 | -6.0472 | 276 |
| | C-UCB | -5.1490±1.6835 | -8.1142 | 151 |
| | Softmax | -4.4801±1.1440 | -6.0536 | 433 |
| | TS | -4.4772±0.7694 | -5.7674 | 386 |
| | UCB-E | -4.8374±1.2162 | -6.6388 | 789 |
| | S-Halving | -3.9992±1.1429 | -6.6801 | 193 |
| | Expectation | -4.0141±1.5697 | -6.2238 | 376 |
| | DSEBO (ours) | **-10.6613±2.4279** | **-14.2513** | 294 |