# OpenReview forum: "Automated Random Embedding for Practical Bayesian Optimization with Unknown Effective Dimension"
_ICLR.cc/2026/Conference — ICLR 2026 Conference Withdrawn Submission_

### Official Review · Reviewer_UuUe · 2025-10-27

**Soundness:** 2
**Presentation:** 3
**Contribution:** 2
**Rating:** 2
**Confidence:** 5

**Summary:**

This paper addresses the problem of high-dimensional black-box optimization. In black-box optimization, one aims to either minimize or maximize a function that can only be observed point-wise, and the cost of observing is either slow or incurs a cost. One popular technique for such functions is Bayesian optimization, which uses a probabilistic surrogate of the objective function. The probabilistic surrogate, often a Gaussian process, requires exponentially more data as the dimensionality increases linearly, rendering high-dimensional black-box optimization an inherently difficult problem.

This paper builds on a popular line of research that uses random embeddings to optimize the objective function. The idea behind random embeddings is that if the objective function possesses a low-dimensional effective subspace that, once it is identified, is sufficient to optimize it, one can model the surrogate in the effective subspace. Whenever one needs to evaluate the function, one projects the point to the full-dimensional search space and adds the low-dimensional representation of the next candidate point alongside its associated function value to the low-dimensional surrogate model.

Since the effective subspace is unknown, one instead chooses a low-dimensional subspace and an associated projection to the full-dimensional input space. If one possesses no prior knowledge about a new problem, one often resorts to random projections. The challenge then lies in choosing a subspace dimensionality and projection that maximizes the chances of being able to represent the global optimum of the objective.

Addressing the former challenge, this paper proposes a method that grows the dimensionality of the search space over time based on the optimization performance in previous subspaces. Unlike related approaches that a priori assign an evaluation budget to each subspace, the proposed method (DSEBO) ramps up the dimensionality faster if the dimensionality-scaled difference between the best values in the last two subspace is large, and slower otherwise. The slope is clipped to the range of [0.5, 1.5] to avoid stagnation and a sufficiently large dimensionality growth.

The authors compare their method against several other techniques for high-dimensional Bayesian optimization, highlighting the benefits of their approach.

**Strengths:**

- The problem is highly relevant. Subspace-BO methods provide a principled approach for problems satisfying the assumption of an effective subspace but are restricted by the need to guess the effective dimensionality, which usually is infeasible due to the black-box nature of $f$.
- The paper is the first work that proposes a data-driven approach for ``learning'' the effective dimensionality.
- The empirical evaluation compares against a comprehensive set of baselines.

**Weaknesses:**

- The main weakness is the limited empirical evaluation. While the set of baselines is relatively large, the set of benchmark problems is relatively small. For instance, the authors only run $d_e=30$ for the synthetic problems. Given that the proposed algorithm is supposed to adapt to different effective dimensionalities, it feels like a minimal requirement that the empirical evaluation runs different effective dimensionalities and studies the behavior of the algorithm when doing so. Also, the set of real-world experiments is limited. See, e.g., BAxUS for a more comprehensive empirical evaluation.
- The overall approach feels relatively heuristic. For instance, the slope is always at least 0.5 so that the expansion never stops. I assume that this is motivated by empirical results.
- Subsections 4.1 and 4.2 explain the shared embedding. This is not a novel idea. BAxUS [2] used a very similar construction, but it is not cited appropriately here. Readers could get the impression that this is an entirely novel contribution. I am wondering why you do not adopt the BAxUS embedding since it avoids projecting outside the search space.

**Questions:**

- For a comparison against recent state-of-the-art methods, can you compare against Hvarfner, C., Hellsten, E. O., & Nardi, L. (2024, July). Vanilla Bayesian Optimization Performs Great in High Dimensions.?
- Why not use a HeSBO-type projection matrix like Papenmeier et al., 2022?
- In line 285, you write that your approach has a lower computational cost. Since the cost of GP inference is mainly dictated by the number of function evaluations, can you clarify in how far your approach lowers computational cost?
- In the same sentence, you state that unlike BAxUS, DSEBO only expands the subspace dimension in within the range $[d_l,d_h]$. This is true, but how do you set $d_h$? If you set it to $D$, you also eventually reach full input dimensionality. Otherwise, you need to make an uninformed guess.

---

### Official Review · Reviewer_mBgf · 2025-11-01

**Soundness:** 2
**Presentation:** 2
**Contribution:** 2
**Rating:** 4
**Confidence:** 3

**Summary:**

This paper proposes DSEBO for high-dimensional black-box optimization with unknown effective dimension. DSEBO combines random embedding with an adaptive subspace selection mechanism. The method (i) optimizes in a low-dimensional random embedding, (ii) expands the subspace dimension when a convergence criterion is met, and (iii) shares a single embedding matrix across subspaces so data collected in smaller subspaces remain reusable after expansion. Experiments on crafted high‑D synthetic functions (D∈{1k,10k}) and three “real‑world” oracles (MSLR, Lasso‑Hard, LIMO) show strong best‑so‑far curves versus embedding and non‑embedding baselines

**Strengths:**

Clear algorithmic framing with reusable data via shared embeddings: Figure 1 illustrates a clear algorithmic framework that utilizes shared embeddings for data reuse. This is achieved by generating subspace matrices from a single source, specifically by taking the first 'd' rows, which allows for efficient data reuse after expansion (as detailed in Section 4.2 and Figure 1). This approach is technically sound and particularly beneficial for budget-constrained Bayesian Optimization. The provided pseudo-code further specifies the necessary inputs and the expansion schedule parameter.
Theoretical Novelty: The paper provides a theoretical analysis with regret bounds for DSEBO, which is a non-trivial extension of GP-UCB theory to this multi-phase optimization setting. Prior high-dimensional BO works seldom include rigorous regret bounds for adaptive subspace methods. DSEBO’s analysis highlights the trade-off between approximation error (from projecting into a smaller subspace) and optimization error (from searching a large space with limited samples). The authors derive a simple regret bound.
Baselines include prominent high-D methods (REMBO, HesBO, ALEBO, BAxUS, TuRBO, etc.) support comparative claims.

**Weaknesses:**

Weakness:
Limited statistical analysis: Results are mean ± standard error over 10 runs, but there is no mention of significance testing (e.g., Wilcoxon, t-tests) or rank aggregation across tasks.
Missing real-world problems: Showing success on 6 synthetic problems does not guarantee success on real-world problems. MSLR and LIMO are real-world tasks. However, Lasso-Hard is a synthetic problem in LassoBench. This paper should consider benchmarking the actual real-world problems in LassoBench (e.g., Lasso Breast_cancer, Diabetes, Leukemia, DNA, RCV)
Clarity and Reproducibility: The paper suffers from poor clarity and significant reproducibility gaps, despite the author providing code. Several crucial experimental details are either missing or inconsistently reported. Specifically, the number of trials, seeding policy, initial design size, evaluation budgets per task, and consistent hardware/runtime information are unclear or absent.


Novelty: The novelty relative to prior structured random-embedding approaches (REMBO, BAxUS) could be more rigorously positioned. While useful, the core idea of dynamic subspace expansion is not entirely new as BAxUS already proposed using a nested sequence of subspaces that grow towards the full dimension. The adaptive schedule for increasing dimension, albeit novel in heuristic form, lacks a principled derivation and feels ad-hoc. It is inspired by observing improvement slopes, but it’s not clear if this is significantly better than a simpler schedule (e.g., always increase by a fixed amount or double the dimension). The paper does not compare against BAxUS on equal footing in a dedicated experiment to isolate this difference – we only see that DSEBO outperforms BAxUS overall, but it’s unclear how much is due to the specific dynamic strategy versus other factors (such as different GP hyperparameter settings or implementation details).

**Questions:**

Ablation on expansion policy: How much of the gain comes from the dynamic k-scaled step (Algorithm 1, line 9) vs. any monotone schedule?

Insufficient experiment: The paper mentioned “All algorithms are independently repeated 10 times.” Only 10 independent runs is not enough. BAxUS and Standard GP, for example, repeat 20 times per algorithm per run.


Initialization samples generation is ambiguous: The paper mentioned “In the first iteration t=1, the dataset is empty and initialized with a random point.” What random? Is it random, Sobol, or Latin hypercube? Is the set of initialized samples fixed for all algorithms? If not, none of the results will make sense due to the failure of control experiments, since initial samples will affect performance. (See: https://doi.org/10.1115/1.4063006)
Seed policy not stated: The current policy on random seeds is unclear. It is crucial to document and publish the specific random seeds used, especially when they are shared across different methods within a given run. Given the substantial impact of random seeds on Bayesian Optimization (BO) outcomes, statistical testing is essential to ensure the reliability of results.
Unclear selection of benchmark problems: Is there a reason for the selection of these sets of benchmarks? Other papers in the citations have tested on this set of synthetic benchmarks: Ackley, Powell, Rastrigin, and StyblinTang, and also these real-world problems: Mopta08, Rover, LassoBench’s actual real-world problems, SVM, Mujoco robotic problems. If the paper does not compare to these commonly used real-world benchmarks, the comparison is unfair. Also, the Lasso-Hard is a synthetic problem based on the original LassoBench paper.
Convergence plot’s y-axis: Please clarify if "best-so-far" is equivalent to "regret." If so, all plots should be updated to maintain consistent terminology, as the paper discusses regret but does not explicitly plot it.
Subtle differences in performance: In Figure 3, we see that DSEBO has very similar convergence to other baseline algorithms in Rosenbrock and Griewank, as well as in Figure 5’s MSLR. It is not clear which is better because the wins are very close. Even in the tables in the appendix, some of the differences are down to the second decimal (Table 2 MSLR). It remains unclear what the margin of (dis)advantage is. Does it outperform other methods by a lot, or is the performance difference negligible?
Add statistical rigor: A statistical ranking (e.g., Wilcoxon, Friedman) of algorithms will make the paper's case for the advantages of methods across problems. See how these papers perform ranking: https://arxiv.org/pdf/1809.04356, https://arxiv.org/pdf/2305.17535
Scalability not demonstrated: The paper claims better scalability via dynamic subspace growth and mentions improvements “in time,” but concrete scalability plots/tables are not present in the visible sections. Also, showing the result only on D=1000 and 10000 is not necessarily a scalability proof. Scalability means to scale from low-dimensional to high-dimensional problems. What about the dimensions in between? How about the lower-dimensional problems?
Limitations of this method: Aren’t there any limitations of DSEBO? Please elaborate.
Please report per‑iteration runtime and total wall‑clock vs. BAxUS and TuRBO on at least one synthetic and one real task.
Inconsistent iteration across methods: I understand there are computational limitations, but it is unfair to run, for example, SAASBO, ALEBO, and VAEBO with fewer iterations than your method, since they also appear to be still converging in Figures 3 and 4. Why not run it longer for a fair comparison?
Inconsistency between the code and the paper: In Section 4.2 and Figure 2, the paper claims that “the shared embedding process across different subspaces” is formed by “appending zeros”. However, the code provided uses:

train_Xs_sample = embedder.dimProjection(dLast, currentDim, train_Xs_sample)

and def dimProjection(self, dimLow, dimHigh, x):
        if dimHigh > self.maxOptDim or dimLow >= dimHigh:
            raise ValueError('Invalid dimension')
        matLow = self.randomMatrix[:, :dimLow]
        matHighInv = torch.pinverse(self.randomMatrix[:, :dimHigh])
        obj_x = torch.matmul(matHighInv, torch.matmul(matLow, x.t())).t()
        return obj_x

Which computes z’, that does not maintain exact equality to f stated in section 4.2. I also did not find the zero padding in the code.

---

### Official Review · Reviewer_uVne · 2025-11-01

**Soundness:** 3
**Presentation:** 2
**Contribution:** 2
**Rating:** 2
**Confidence:** 3

**Summary:**

The paper introduces a method for high-dimensional Bayesian optimization based on random embeddings. The particular approach taken here is to begin with a low-dimensional random embedding, and then allow it to expand and add new features whenever the optimization is getting stuck in the current embedding.

The paper includes a regret bound, and evaluates performance on several benchmarks.

**Strengths:**

* The paper offers some new results and strategies for using random embeddings for Bayesian optimization. The approach is well-motivated and uses reasonable heuristics for determining when to add new dimensions.

* The empirical evaluation is very thorough and includes all of the baselines I'd want to see, and then some.

* The theoretical analysis was nice to see.

**Weaknesses:**

The main challenge for this paper is that the core idea, starting from a small random subspace and adding other dimensions as needed, was already done in BAxUS (Papenmeier et al. 2022).

Because the core idea of the paper is already published and the difference is a matter of how that idea is implemented, the paper badly needs a very clear and detailed description of the full set of differences between BAxUS and the new method, DSEBO. Currently the only discussion of the difference between BAxUS and DESBO is that BAxUS "increases dimensions exponentially and eventually optimizes in the full space, DSEBO adopts the proposed dynamic dimension expanding strategy. Specifically, DSEBO automatically and nearly linearly expands the subspace dimension within range [dl, dh], leading to lower computational costs and better scalability."

To the extent that the difference between the methods is just the rate at which new dimensions are added, this does not seem to me a significant enough contribution for a conference like ICLR.

The paper would also benefit from editing for grammar.

**Questions:**

Could you provide a detailed set of differences between BAxUS and DSEBO?

---

### Official Review · Reviewer_zTzC · 2025-11-01

**Soundness:** 3
**Presentation:** 3
**Contribution:** 3
**Rating:** 6
**Confidence:** 1

**Summary:**

Traditional methods use fixed subspace dimensions provided by experts or rely on trial and error to estimate subspace dimensions with many resources consumed. DSEBO dynamically determines the dimension of the next subspace based on the quality of the solutions in different subspaces and shares the queried solutions with the new subspace to achieve a better initialization.

**Strengths:**

1. Clearly written
2. Experiments is extensive

**Weaknesses:**

1. The ablation study lacks the analysis on the scaling of the dimension. Since the most of the experments is done at D=1000.

**Questions:**

1. Why the best-so-far are different when the iterations is 0?

---

### Note · Authors · 2025-11-12

I have read and agree with the venue's withdrawal policy on behalf of myself and my co-authors.